



**The variation and visualisation of elastic anisotropy in rock forming minerals**
David Healy[1], Nicholas E. Timms[2] & Mark A. Pearce[3]
[1]: School of Geosciences, King's College, University of Aberdeen, Aberdeen AB24 3UE, United
Kingdom
[2]: Space Science and Technology Centre, School of Earth and Planetary Sciences, Curtin
University, Perth, GPO Box U1987, WA 6845, Australia
[3]: CSIRO Mineral Resources, Australian Resources Research Centre, 26 Dick Perry Avenue,
Kensington, WA 6151, Australia
*corresponding author: d.healy@abdn.ac.uk
**Abstract**
All minerals behave elastically, a rheological property that controls their ability to support stress,
strain and pressure, the nature of acoustic wave propagation and influences subsequent plastic (i.e.
permanent, non-reversible) deformation. All minerals are intrinsically anisotropic in their elastic
properties – that is, they have directional variations that are related to the configuration of the
crystal lattice. This means that the commonly used mechanical elastic properties that relate elastic
stress to elastic strain, including Young's modulus ($E$), Poisson's ratio ($\nu$), shear modulus ($G$) and
linear compressibility ($\beta$), are dependent on crystallographic direction. In this paper, we explore the
ranges of anisotropy of $E$, $\nu$, $G$ and $\beta$ in 86 rock-forming minerals, using previously published data,
and show that the range is much wider than commonly assumed. We also explore how these
variations (the directionality and the magnitude) are important for fundamental processes in the
solid earth, including deformation (mechanical) twinning, coherent phase transformations and
brittle failure. We present a new open source software package (AnisoVis, written in MATLAB),
which we use to calculate and visualise directional variations in elastic properties of rock-forming
minerals. Following previous work in the fields of chemistry and materials, we demonstrate that by
visualising the variations in elasticity, we discover previously unreported properties of rock forming
minerals. For example, we show previously unreported directions of negative Poisson's ratio and
negative linear compressibility and we show that the existence of these features is more widespread
(i.e. present in many more minerals) than previously thought. We illustrate the consequences of
intrinsic elastic anisotropy for the elastic normal and shear strains within $\alpha$-quartz single crystal
under different applied stress fields; the role of elastic anisotropy on Dauphiné twinning and the $\alpha$-
$\beta$ phase transformations in quartz; and stress distributions around voids of different shapes in talc,
lizardite, albite, and sanidine. In addition to our specific examples, elastic anisotropy in rock-
forming minerals to the degree that we describe has significant consequences for seismic (acoustic)
anisotropy, the focal mechanisms of earthquakes in anisotropic source regions (e.g. subducting
slabs), for a range of brittle and ductile deformation mechanisms in minerals, and geobarometry
using mineral inclusions.



## Introduction

The elastic deformation of rock-forming minerals plays an important role in many earth processes.
The increased availability of measured or calculated elastic properties of whole rocks and of
specific rock-forming minerals has led to advances in many fields of earth science, including
seismology, geodynamics, tectonics and metamorphism. Minerals have long been known to display
anisotropy – directional variations – in their elastic properties (Mandell, 1927; Birch & Dancroft,
1938; Hearmon, 1946), and that these variations show a close relationship to the symmetry of the
mineral crystallographic structure. Advances in laboratory methods of measurement (acoustic
velocities, Brillouin scattering, resonant ultrasound) and in theoretical techniques for *ab initio*
calculations has allowed scientists to quantify this anisotropy for a wide range of rock forming
minerals. For this paper we have collected 246 published datasets (measurements or *ab initio*
calculations) of anisotropic elastic properties covering 86 distinct minerals. Elastic anisotropy is
fully described by a fourth rank tensor (compliance or stiffness, see below), and published data are
commonly presented in a Voigt matrix format, listing up to 21 independent values (depending on
the crystal symmetry class), whereas elastically isotropic minerals require only 2 independent
values. A key aim of this paper is to use published data to visualise and explore elastic anisotropy
in rock forming minerals using familiar measures, such as Young's modulus and Poisson's ratio,
but presented in novel formats and thereby render the increasing volume of data more transparent to
analysis. As noted by previous authors (Karki & Chennamsetty, 2006; Lethbridge et al., 2010;
Marmier et al., 2010; Gaillac et al., 2016), graphical depictions of the directional variation of elastic
properties provide new opportunities to relate the quantitative data to the crystalline structure of the
mineral. This in turn allows us to relate the observed or predicted mechanical and chemical
behaviour of the mineral to specific crystallographic directions.
It has long been recognized that the velocity of seismic waves passing through rocks is a direct
function of the minerals' elastic properties and their density, expressed through the Christoffel
equation (Christoffel, 1877; Zhou & Greenhalgh, 2004). By considering rocks as polycrystalline
aggregates various workers have modelled seismic velocities, and their anisotropy, by combining
single mineral elasticity data with different averaging schemes due to Reuss, Voigt or Hill (e.g.
Mainprice, 1990; Lloyd & Kendall, 2005). This 'rock recipe' approach has improved our
understanding of the composition and structure of the lower crust and mantle and provided useful
constraints for alternative models for observed variations in seismic anisotropy beneath continents
and around arcs (e.g. Kern, 1982; Tatham et al., 2008; Healy et al., 2009).
Inclusions of one mineral or fluid within another host mineral have been used to estimate pressures
at the time of inclusion or entrapment (Rosenfeld & Chase, 1961; Rosenfeld, 1969; Chopin, 1984;
Gillet et al., 1984; van der Molen & van Roermund, 1986; Angel et al., 2014; Angel et al., 2015).
The analysis critically depends on the elastic properties of the host mineral and, in the case of solid
inclusions, of the inclusion itself, typically expressed as the bulk and shear moduli (e.g.
Mazzucchelli et al., 2018). The underlying theory is based on the classical analysis by Eshelby
(1957, 1959) who derived the equations for the deformation within an ellipsoidal inclusion and host
due to the imposition of a far-field load. Most of the work to date has simplified the analysis to
assume isotropy in both the inclusion and the host, although see Zhang (1998) for a rare exception.
Therefore, the full effects of host minerals and inclusion elastic anisotropy on inclusion-based
geobarometry have not yet been rigorously investigated. Furthermore, fluid inclusions can
decrepitate – i.e. fracture their host and dissipate their fluid – if their internal overpressure rises to a



critical value that exceeds the local strength of the enclosing grain. The basis for predicting this
behaviour is linear elastic fracture mechanics (LEFM), and the assumption of elastic isotropy is
nearly ubiquitous (e.g. Lacazette, 1990).
Permanent, non-reversible (i.e. plastic) deformation of minerals is invariably preceded by an elastic
response prior to some form of yield condition being reached. For example, the elastic properties of
minerals are important in the analysis of brittle cracking at the grain scale. As noted above for the
decrepitation of fluid inclusions, the dominant paradigm for this analysis is linear elastic fracture
mechanics (LEFM), and the assumption of elastic isotropy. This is important because faults and
fractures in rocks are composite structures, built by the interaction and coalescence of many smaller
cracks that nucleate at the scale of individual grains i.e. within elastically anisotropic crystals.
Jaeger & Cook (1969) used the equations published by Green & Taylor (1939) to consider the
stresses developed at the edges of circular holes in anisotropic rocks. In their analysis (repeated in
Pollard & Fletcher, 2005), they dismissed the significance of elastic anisotropy because the ratio of
maximum to minimum Young's modulus in rocks is 'rarely as high as 2'. Timms et al. (2010)
conducted novel indentation experiments in a single crystal of quartz and produced a type of cone
fracture with variations in opening angle and crack length that have a trigonal symmetry radiating
from the point of contact, and thus demonstrated the key role played by the elastic anisotropy in
controlling the fracture geometry. In the same study, these authors confirmed that elastic
anisotropy plays a significant role in controlling the focal mechanisms (moment tensors) of acoustic
emission events at the scale of a single crystal.
Poisson's ratio appears as a term in, for example, the equations describing fracture toughness and
indentation, and therefore the precise value of Poisson's ratio is important. Poisson's ratio for
isotropic materials is constrained to lie between 0.5 and −1, but there are no theoretical limits for
anisotropic materials (Ting & Chen, 2005). Materials with Poisson's ratio less than 0 are termed
'auxetic' (Lakes, 1987; Baughman et al., 1998a; Prawoto, 2012; Pasternak & Dyskin, 2012).
Fracture toughness and resistance to indentation increase as Poisson's ratio approaches the lower
(isotropic) limit of −1.0 (Yeganeh-Haeri et al., 1992). In rock forming minerals, negative Poisson's
ratios have already been documented for $\alpha$-cristobalite (Yeganeh-Haeri et al., 1992), for quartz at
the $\alpha$-$\beta$ phase transition (Mainprice & Casey, 1990), for talc (Mainprice et al., 2008), and for calcite
and aragonite (Aouni & Wheeler, 2008). A key question therefore is to determine if there are other
rock forming minerals with the same properties, and for which specific crystallographic directions.
In a recent review of data on Poisson's ratio in engineering materials, Greaves et al. (2011) pointed
out that the brittle-ductile transition at the grain scale is also a function of the elastic properties and
therefore likely dependent on direction in strongly anisotropic materials.
Elastic properties, and anisotropy, is also known to influence the 'ductile' or plastic deformation of
minerals, and has a role in twinning, crystal plasticity (dislocation creep) and phase transformations
(e.g. Tullis, 1969; Christian & Mahajan, 1995; Timms et al., 2018). The role of mineral elasticity is
also important for inhomogeneous distribution of stresses at the grain scale necessary for driving
pressure solution creep, and is either treated implicitly (e.g., Wheeler, 1992) or explicitly (e.g.,
Wheeler, 2018). However, in many studies of rock deformation, minerals are commonly assumed to
be elastically isotropic and scalar mean values of elastic moduli are used, and/or elastic strains are
assumed to be small relative to plastic deformation and so ignored (e.g., in visco-plastic self-
consistent (VPSC) code) (Tomé & Lebensohn, 2014).

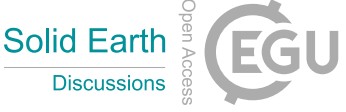



Given the key role that the elastic behavior of minerals plays in so many fundamental geological
processes, the scientific need to explore, understand and quantify directional variations in elastic
properties in minerals is clear, as is the need to develop better approaches to their graphical
visualisation. It is very difficult to full appreciate the variations in elastic properties of a mineral
simply by inspection of the 4$^{th}$ rank stiffness (or compliance) tensor, even in reduced form (Voigt
notation; see below). A related requirement is the ability to investigate the interactions of mineral
elastic anisotropy with imposed pressure, stress, or strain. However, the visualisation and full
appreciation of the properties of 2$^{nd}$ rank tensors, such as stress and strain, also presents challenges.
No single surface can simultaneously portray the full anisotropy quantified by the diagonal (normal)
and off-diagonal (shear) components of these tensorial mechanical quantities. Depictions of strain
(or stress) as ellipsoids using only the principal values as semi-axes fail to quantify the directional
variations in shear strain (or stress) and cannot easily show examples with mixed positive and
negative principal values. We believe there are clear educational benefits to alternative approaches
to visualising stress and strain, which students commonly find challenging, both conceptually and
from a 3-dimensional cognition perspective. For example, most geological textbooks either
illustrate stress or strain as ellipses/ellipsoids of the normal component only (with the limitations
described above), Mohr diagrams, or written out in matrix notation. Furthermore, a common
misnomer that some minerals are isotropic in material properties undoubtedly stems from the strong
emphasis on optical properties of minerals in most undergraduate mineralogy courses. Software
tools with the capability of comparative visualisation of various physical properties of minerals in
2- and 3-dimensions, including elastic, optical, and acoustic anisotropy have a valuable place in
teaching and learning in mineralogy and in scientific research.
While the number of published datasets for single mineral elastic anisotropy continues to increase,
there are relatively few publications that have reviewed or synthesised the available data. Gercek
(2007) provided a useful review of Poisson's ratio for rocks and included some data for specific
minerals. A more recent review of Poisson's ratio in rocks (Ji et al., 2018) also contained data for
minerals, but used their calculated Voigt-Reuss-Hill average values rather than quantify their
anisotropy. Workers in the fields of chemistry, physics and engineering have published methods
and tools for visualising the elastic anisotropy of various groups of solid elements and compounds
(Karki & Chennamsetty, 2006; Lethbridge et al., 2010; Marmier et al., 2010; Gaillac et al., 2016),
and these predominantly focus on Poisson's ratio. In earth sciences, the MTEX toolbox for the
analysis and modelling of crystallographic textures from electron backscatter diffraction (EBSD)
data provides stereographic projections of elastic properties, such as Young's modulus, for single
minerals (Hielscher, R. & Schaeben, H., 2008; Mainprice et al., 2011). The MSAT toolbox for
seismic anisotropy also contains options for plotting the elastic anisotropy of rocks and minerals
(Walker & Wookey, 2012). Both MTEX and MSAT provide one or more options for displaying
the elastic properties of minerals, but their main focus is on the analysis of textures and seismic
(acoustic) velocity anisotropy, respectively.
In this paper we present the AnisoVis toolbox, a collection of new MATLAB scripts based on
published methods with a graphical user interface (GUI), to explore the range of elastic anisotropy
displayed by rock forming minerals. Specifically, AnisoVis depicts the magnitude of the
directional variations in elastic properties such as Young's modulus ($E$), Poisson's ratio ($\nu$), shear
modulus ($G$) and linear compressibility ($\beta$) using a range of 2- and 3-dimensional representations of
each elastic property to enable a complete assessment of the anisotropy in relation to the crystal





symmetry. We exploit the large database of published elastic constants for rock-forming minerals
to systematically assess the anisotropy of different elastic properties as a function of temperature
and pressure (where possible), giving new insights into the elastic behaviour of rock-forming
minerals. Most of the figures presented in this paper have been produced from the AnisoVis
toolbox, which is freely available on the web.
A table of symbols and terms used in this paper is provided in Table 1. We follow the geological
convention that compressive stress is positive, tensile stress is negative. Elastic properties are
reported in SI units. In Section 2 we review the theoretical basis of linear elasticity and the formal
description of elastic anisotropy in terms of the key equations. We then describe the methods we
use to visualise and quantify the directional variations in elastic properties for any given mineral.
We present two sets of results. Firstly, we analyse general trends in the database of 86 distinct
minerals with 246 separate elasticity datasets from published sources, and summarise the degree of
anisotropy to be found in rock forming minerals. Secondly, we analyse specific examples and focus
on their response to applied deformation. We review the key issues raised by these analyses in the
Summary. The Appendix contains benchmarks of the calculations performed in AnisoVis in
comparison to published output from previous workers.

| Quantity | Symbol | Default SI unit |
|---|---|---|
| Young's modulus | $E$ | Pa |
| Poisson's ratio | $\nu$ | |
| Shear modulus | $G$ | Pa |
| Linear compressibility | $\beta$ | $Pa^{-1}$ |
| Bulk modulus | $K$ | Pa |
| Compliance | $s$ | $Pa^{-1}$ |
| Stiffness | $c$ | Pa |
| Stress | $\sigma$ | Pa |
| Strain | $\varepsilon$ | |
| Normal stress | $\sigma_n$ | Pa |
| Shear stress | $\tau$ | Pa |
| Normal strain | $\varepsilon_n$ | |
| Shear strain | $\gamma$ | |
| Unit vectors parallel to crystallographic axes | $\boldsymbol{a, b, c}$ | Miller notation |

**Table 1.** List of symbols and terms used in this paper, together with their default units (if any).






## 2. Theory and underlying equations

The elastic anisotropy of a solid material is described by a fourth rank tensor, either the compliance
$s_{ijkl}$ or its inverse, the stiffness $c_{ijkl}$. For linear elastic deformation, the generalised form of Hooke's
Law can be written as:
$\qquad \varepsilon_{ij} = s_{ijkl}\,\sigma_{kl}$ (1)
where $\varepsilon_{ij}$ and $\sigma_{ij}$ are the second rank tensors of strain and stress, respectively. Alternatively,
equation (1) can be written as:
$\qquad \sigma_{ij} = c_{ijkl}\,\varepsilon_{kl}$ (2).
Symmetry considerations lead to $s_{ijkl} = s_{ijlk}$ and $s_{ijkl} = s_{jikl}$ (Nye, 1985). The corollary of these
relationships is that the number of independent (potentially unique) components of $s_{ijkl}$ is reduced
from 81 (=$3^4$) to 36. The same applies to $c_{ijkl}$. The elastic compliance $s$ or stiffness $c$ of a crystal
can therefore be represented in a more compact form, known as the Voigt matrix. This is a square 6
x 6 matrix where, for example, the elements of elastic stiffness are defined as $c_{IJ} = c_{ijkl}$, where $I = ij$
and $J = kl$. There are six different permutations of $I(J) = ij(kl)$, the details of which are listed in
Nye (1985) and more recently in Almqvist & Mainprice (2017).
The measured and calculated elastic properties of single crystals are reported in Voigt matrix
notation ($s_{IJ}$, $c_{IJ}$), where the indices $I$, $J$ (=1,2,3) relate to a standard Cartesian reference frame ($x$=1,
$y$=2, $z$=3). The relationship between any specific crystal lattice and this Cartesian reference is
arbitrary, but we adopt the convention described in Britton et al. (2016). In this system:
• the unit cell lattice vectors $a$, $b$, and $c$ form a right-handed set,
• $c$ is parallel to Cartesian $z$,
• $b$ lies in the Cartesian $y$-$z$ plane at angle $\alpha$ to $c$, and
• $a$ is directed at angle $\beta$ to $c$ and $\gamma$ to $b$.
Note that $\alpha$ is the angle between $b$ and $c$, $\beta$ is the angle between $c$ and $a$ and $\gamma$ is the angle between
$a$ and $b$ (see Figure 1a).



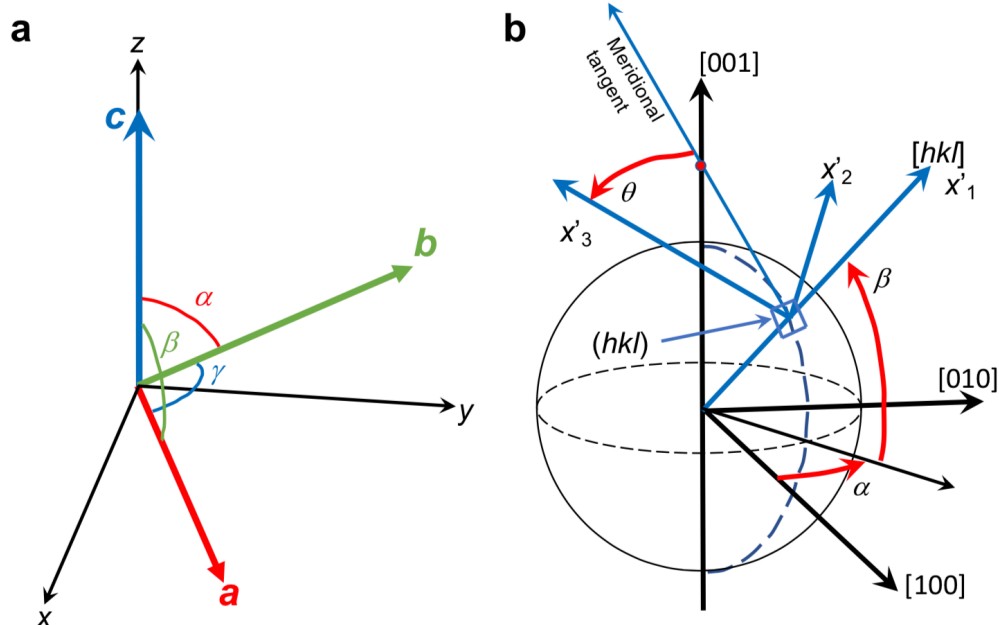


**Figure 1. a**) Crystallographic orientation convention (after Britton et al., 2016) and **b**) geometrical
reference frame (after Turley & Sines, 1971) used in this paper.

Familiar elastic properties, such as Young's modulus ($E$), Poisson's ratio ($\nu$) and shear modulus
($G$), can be expressed directly in terms of the components of the compliance matrix. For example,
the Young's modulus of a single crystal for a uniaxial stress applied in the $x$-direction is:

$$E_x = E_I = 1 / s_{11} \qquad (3)$$

and the Poisson's ratio for a uniaxial stress and axial strain along $x$ and a lateral strain along $y$ is

$$\nu_{xy} = -s_{21} / s_{11} \qquad (4)$$

Note that, in general for anisotropic materials, $\nu_{xy} \neq \nu_{yx}$ etc.

Guo & Wheeler (2006) note that although Poisson's ratio may be negative for some directions,
these are often compensated by higher positive values in transverse directions perpendicular the
minima in the same plane. They suggest a more useful measure of extreme auxeticity, the **areal
Poisson's ratio**, defined as the average of all values of Poisson's ratio taken within the plane
normal to a chosen direction. If the areal Poisson's ratio is negative this implies that a cylinder of
the mineral would contract under a uniaxial compression, around the whole circumference, and not
just along certain directions.

In order to calculate specific values of these elastic properties in more general directions within a
crystal – i.e. not just along the axes of the default Cartesian reference frame – we need to transform
the compliance matrix into a different reference frame. We follow the notation used by Turley &
Sines (1971) based on Eulerian angles $\alpha$, $\beta$ and $\theta$ (see Figure 1b) that define the new Cartesian axes





(1', 2', 3' or *x'*, *y'*, *z'*) in relation to the initial reference frame (1,2,3 or *x*, *y*, *z*). The transformation
of compliance matrix $s_{ijkl}$ to $s'_{ijkl}$ is given by (Nye, 1985):

$s'_{ijkl} = a_{im}\, a_{jn}\, a_{ko}\, a_{ip}\, s_{mnop}$          (5)

where the elements of the rotation matrix ***a*** are given by:

$a_{ij} = \begin{bmatrix} A & B & C \\ (D\sin\theta + E\cos\theta) & (F\sin\theta + G\cos\theta) & H\sin\theta \\ (D\cos\theta - E\sin\theta) & (F\cos\theta - G\sin\theta) & H\cos\theta \end{bmatrix}$          (6)

where $A = \cos\alpha\cos\beta$, $B = \sin\alpha\cos\beta$, $C = \sin\beta$, $D = -\cos\alpha\sin\beta$, $E = -\sin\alpha$, $F =$
$-\sin\alpha\sin\beta$, $G = \cos\alpha$, $H = \cos\beta$ (Turley & Sines, 1971).
Using the transformed compliance matrix $s'_{ijkl}$, we can now calculate the elastic properties for any
general direction within the crystal defined by a unit vector with angles $\alpha$, $\beta$ and $\theta$, for example:

$E'_1 = 1 / s'_{11}$          (7)

$G'_{12} = 1 / s'_{66}$          (8)

$v'_{12} = -s'_{21} / s'_{11}$          (9)

To calculate the variation in any elastic property over all possible directions in 3D, we simply need
to vary $\alpha$ and $\beta$ over a unit sphere ($\alpha$: 0-360°, $\beta$: 0-180°) and vary $\theta$ over a unit circle ($\theta$: 0-360°).
*Isotropic approximations of anisotropic elastic properties*
Two useful 'averaging' schemes that can be applied to the full set of anisotropic elastic properties
of polycrystals are those due to Reuss and Voigt (see Hill, 1952). The bulk and shear moduli in the
Voigt scheme are defined as:
$K^V = [(c_{11} + c_{22} + c_{33}) + 2(c_{12} + c_{23} + c_{31})]/9$          (10)
$G^V = [(c_{11} + c_{22} + c_{33}) - (c_{12} + c_{23} + c_{31}) + 3(c_{44} + c_{55} + c_{66})]/15$          (11)
and in the Reuss scheme as:
$K^R = 1/[(s_{11} + s_{22} + s_{33}) + 2(s_{12} + s_{23} + s_{31})]$          (12)
$G^R = 15/[4(s_{11} + s_{22} + s_{33}) - 4(s_{12} + s_{23} + s_{31}) + 3(s_{44} + s_{55} + s_{66})]$          (13)
The Voigt average of any property always exceeds the Reuss average and the 'true' value lies
somewhere in between. The Voigt-Reuss-Hill (VRH) average of a property is defined as the
arithmetic mean of the Voigt and Reuss estimates e.g. $G^{VRH} = (G^V + G^R)/2$. Note that, although only
formally defined for polycrystals and based on averaging over many grains, the Voigt, Reuss and
VRH estimates are in practice useful for single crystals: if we consider a polycrystal made of many
grains all aligned perfectly parallel, then the elastic anisotropy of this polycrystal is identical to that
of the single crystal.



To plot the variations of disparate elastic properties across minerals with widely different
symmetries and anisotropies, we use the Universal Anisotropy Index ($A^U$), of Ranganathan &
Ostoja-Starzewski (2008), defined as:
$A^U = 5 \dfrac{G^V}{G^R} + \dfrac{K^V}{K^R} - 6$    (14)
where $G^V$ and $K^V$ are the Voigt average shear and bulk moduli, respectively; and $G^R$ and $K^R$ are the
Reuss average shear and bulk moduli, respectively.

**3. AnisoVis – program description and visualisation methods**
The visualisations of elastic anisotropy presented in this paper have been prepared using AnisoVis,
a set of custom scripts linked to a graphical user interface (GUI) and written in MATLAB™. This
code is available as an open source project on GitHub (link) and through the MathWorks™
FileExchange server (link). Single mineral elasticity values are supplied as input data, together
with lattice parameters defining the unit cell and symmetry. The code then calculates the
directional variations in elastic properties and produces outputs of the kinds shown in Figures 4-7.
AnisoVis can also calculate the acoustic velocities (phase and group) and their polarisations, and
the optical birefringence from the refractive indices. Over 240 data files for 86 different minerals
are included (from published sources), and a user guide is provided with the software.
*Installation and input file format*
AnisoVis is installed by copying all of the files from the GitHub or Mathworks FileExchange server
into a folder on the user's computer. AnisoVis will run on any computer with MATLAB installed,
including running Windows, Mac OS X or different versions of Linux. After starting MATLAB,
the working folder or directory should be set to the folder containing all of the installed source
code. The application is started by typing 'AnisoVis' in the Command window of the MATLAB
session. There is only one window in AnisoVis (Figure 2). Click 'Browse…' to show the standard
dialog to open an input file of mineral properties. These are stored in formatted tab-delimited
ASCII text files with an extension of '.mdf2' ('mineral data file'). The user guide supplied with the
software has examples for each different mineral symmetry class.



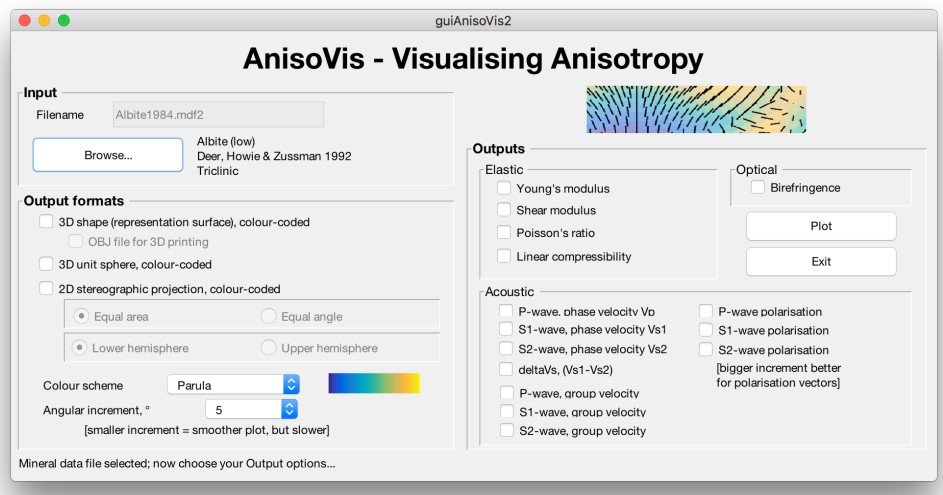


**Figure 2.** The graphical user interface in AnisoVis, showing the range of output options for elastic (and acoustic and optical) anisotropies.

*Calculations*

After selecting the required output formats (shape, sphere or stereogram) and anisotropic properties to be visualised (elastic, acoustic or optical), the user clicks Plot to generate the images. Calculations are performed using the equations for each property described above, looping through three-dimensional space with the specified angular increment. Smaller angular increments (e.g. 1-2°) take longer to run than larger increments (e.g. 5-10°). In the tests that we have conducted to date, run time has been very satisfactory, with most operations completed in a few seconds on standard desktop computers purchased within the last three years. The exception to this performance is when the angular increment is 1°, where run times are typically of the order of 1-2 minutes. We have implemented a MATLAB™ WaitBar to provide basic progress information for lengthier tasks.

*Generating outputs*

Output is directed to MATLAB figure windows, with one plotted property per figure window. These images are automatically saved as '.tif' files at 600 dpi resolution in the working folder. While each figure window is visible, the user can exploit standard MATLAB functionality to resize or reformat the figure as they wish, and can save the figure to a different filename or folder, or even a different graphic format (e.g. '.png' or '.jpeg'). The colour schemes used for the representation surfaces, unit spheres and stereograms can be varied using the drop-down list box in the main window. In addition to the standard MATLAB colour map of 'Parula' we offer 3 other choices from the cmocean colour map library (Thyng et al., 2016) using perceptually uniform scales ('Haline', 'Thermal' and 'Matter').

*Visualising elastic anisotropy in 2-D and 3-D*

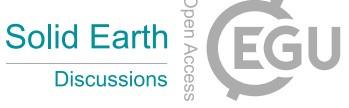



As pointed out by Nye (1985), no *single* surface can represent the elastic behaviour of a crystal
completely. However, we can plot specific surfaces that are useful in practice. To visualise the
anisotropy of elastic properties of single crystals we use a mixture of 3D surfaces and 2D polar
plots projected onto selected planes. We use representation surfaces (Nye, 1985) to generate 3D
shapes where, for any given radius vector measured from the origin to the surface, the radius is
proportional to the magnitude of the property in that direction. The magnitude of the property is
also conveyed by a colour mapping applied to the surface. An alternative method is to plot the
directional variation of a property projected onto a unit sphere, using a colour map to depict the
magnitude. We can also use stereographic projections (lower hemisphere, equal area) to show
directional variations in properties. Lastly, we can use polar plots to the variation of a property in
selected crystallographic planes (e.g. [100], [010], [001]).
*Challenges in visualising Poisson's ratio (ν) and shear modulus G*
Any of the above methods of visualisation can be used for 'simple' elastic properties, such as
Young's modulus or linear compressibility, where the property is a single scalar value for a given
direction. Young's modulus is defined as the ratio of uniaxial stress to uniaxial strain and it is
implicit that the directions of applied stress and measured strain are coincident (i.e. coaxial; Figure
3). However, for Poisson's ratio and shear modulus this is no longer the case. Poisson's ratio is
defined as the ratio of (negative) lateral strain to the axial strain, and therefore involves two
orthogonal directions (Figure 3). Shear modulus is defined as the ratio of the shear stress to the
shear strain, again involving two orthogonal directions (see Figure 3). For a stress (normal or
shear) applied in a specific direction, there is only one value of $E$, but there are many possible
values of $ν$ and $G$. It can be seen from Figure 3 that $ν$ and $G$ will vary according to the direction of
the normal to the chosen direction [*hkl*], described by angle $θ$ in the Turley & Sines (1971) notation.
To plot representation surfaces for $ν$ and $G$, we take their minimum and maximum values calculated
over $θ$ for an applied stress along each direction in 3D-space. In addition, as $ν$ can be negative for
some directions in some minerals, we further separate the minimum representation surfaces of
Poisson's ratio into negative minimum and positive minimum components where appropriate.



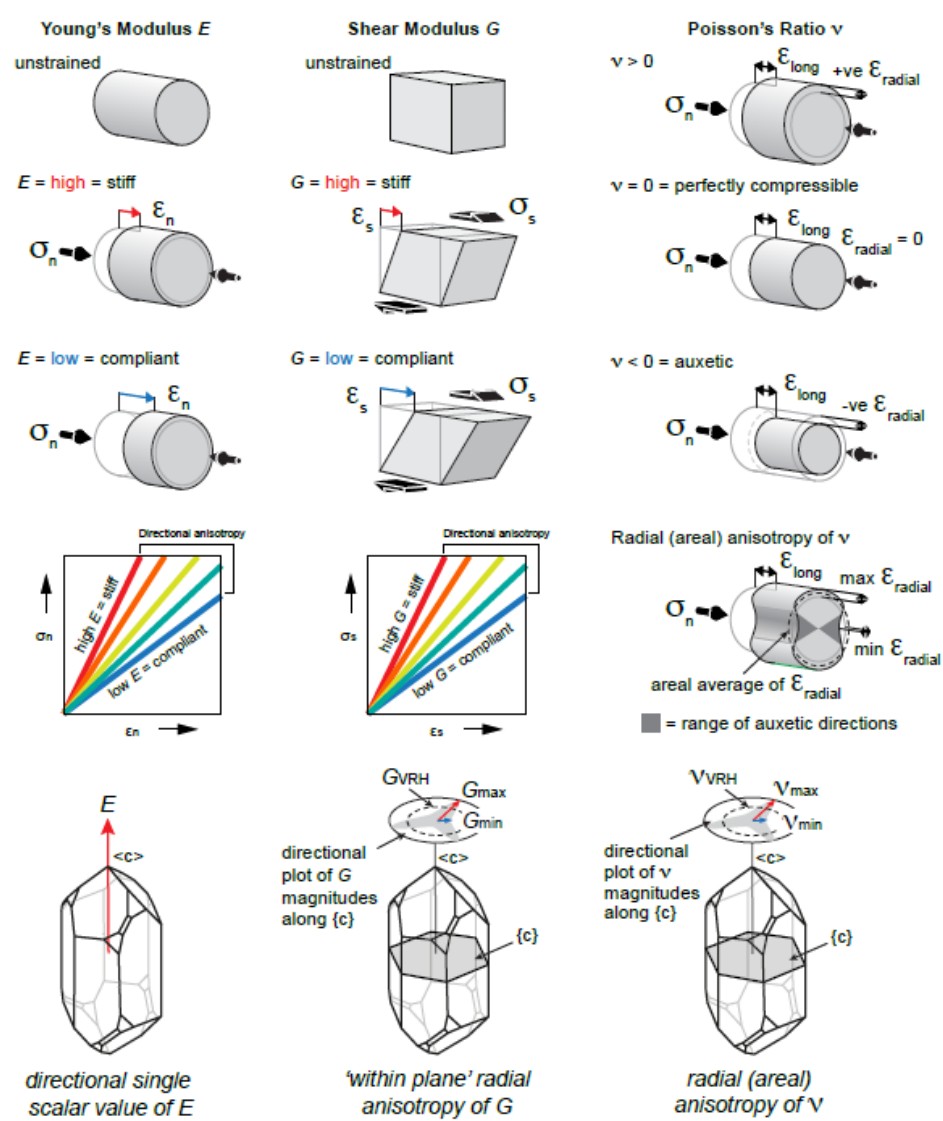


**Figure 3.** Schematic diagrams to illustrate the definitions of Young's modulus, Poisson's ratio, shear modulus in a 3D crystallographic reference frame, using $\alpha$-quartz (trigonal) as an example.

*Example: $\alpha$-quartz (trigonal; Ogi et al., 2006)*

To illustrate the different possibilities described above we use the elasticity of $\alpha$-quartz as quantified by Ogi et al. (2006). The anisotropy of Young's modulus is shown in Figure 4 using a representation surface, a unit sphere, a stereogram and polar plots of E in the plane (100). The colour bar scale is the same in all plots for ease of comparison. Using AnisoVis, the user can rotate any of these plot views in the MATLAB figures to gain a better appreciation of the directional variations in relation to the crystallographic reference axes <a>, <b>, and <c>.


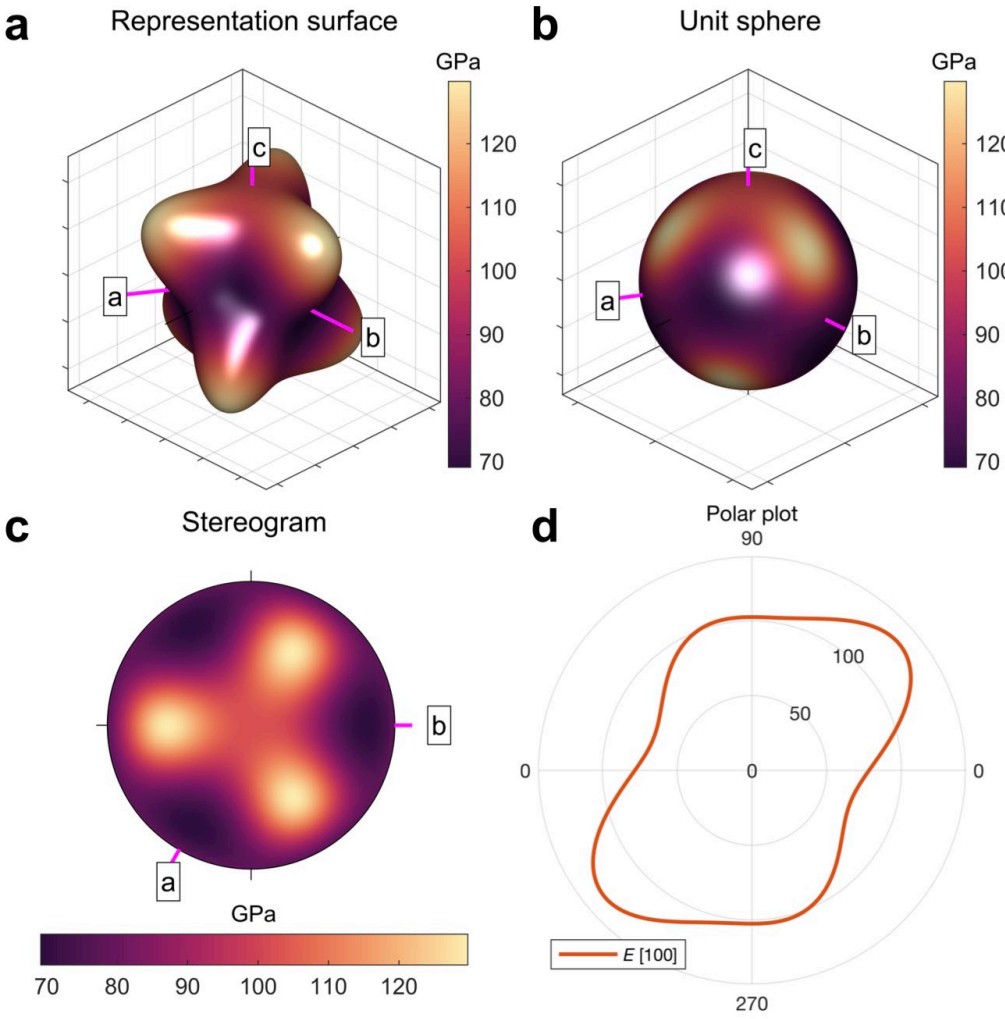

**Figure 4.** Alternative visualisations of the anisotropy of Young's modulus (*E*, in GPa) of α-quartz. **a**) 3D representation surface where the radius in any direction is proportional to the magnitude of *E*. **b**) Projection of *E* on to a unit sphere, colour coded by magnitude. **c**) Lower hemisphere, equal area stereographic projection. **d**) Polar plot of anisotropy of *E* in the [010] plane. Crystallographic axes <a>, <b>, and <c> shown in pink. VRH = Voigt-Reuss-Hill average value of *E*.

As noted above, the shear modulus is a function of shear stress in one direction and a shear strain in a perpendicular direction. Therefore, for any given crystallographic direction in 3D space [*hkl*] in an anisotropic crystal there are many possible values of *G* as the transverse component is rotated through the angle $\theta$ (see Figure 1b). In Figure 5 we show representation surfaces for the minimum and maximum values of *G* of α-quartz associated with each direction [*hkl*]. Polar plots are also shown for (010) and (001).





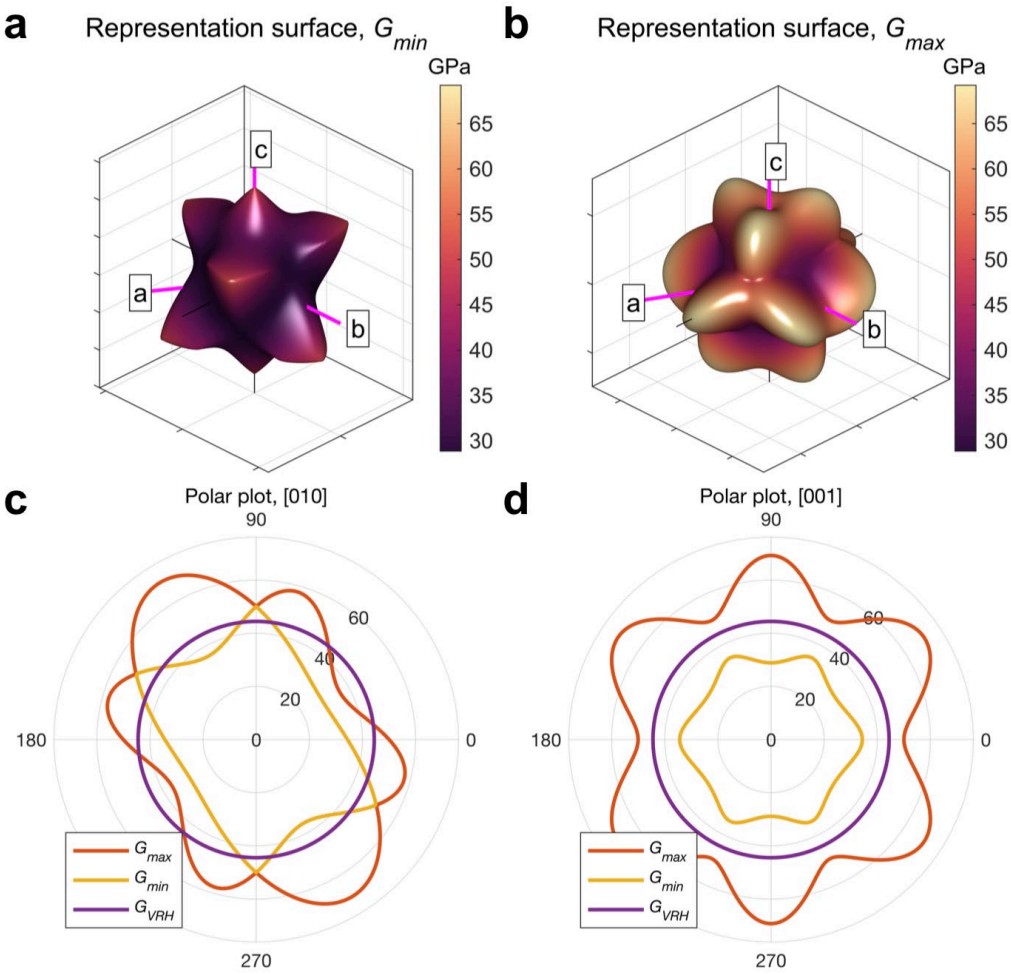

**Figure 5.** Alternative visualisations of the anisotropy of shear modulus ($G$, GPa) of α-quartz. **a-b**) 3D representation surfaces where the radius in any direction is proportional to the magnitude of $G$. Separate surfaces shown for minimum and maximum $G$. **c-d**) Polar plots of anisotropy of $G$ in the [010] and [001] planes, respectively. Crystallographic axes \<a>, \<b>, and \<c> shown in pink. VRH = Voigt-Reuss-Hill average value of $G$.

Visualising the directional variation of Poisson's ratio $\nu$ can pose further challenges. α-quartz is auxetic and has many directions that show negative Poisson's ratios. As for shear modulus, we show representation surfaces for both the minimum (Figure 6a-b) and maximum (Figure 6c) Poisson's ratios, but we separate the minimum Poisson's ratio plot into two surfaces: one for $\nu_{min} < 0$ (Figure 6a) and one for $\nu_{min} > 0$ (Figure 6b). We also include a plot for the areal Poisson's ratio – the value of Poisson's ratio averaged over all $\theta$ for each direction [$hkl$] (Figure 6d, after Guo & Wheeler, 2006). Polar plots for specific 2D planes can also be useful (Figure 6e-f).





**Figure 6.** Alternative visualisations of the anisotropy of Poisson's ratio ($v$) of α-quartz. **a**-d) 3D representation surfaces where the radius in any direction is proportional to the magnitude of $v$. Separate surfaces shown for minimum negative, minimum positive, maximum and areal $n$, as





defined in the equations in Section N.N. **e-f**) Polar plots of anisotropy of $v$ in the [010] and [001]
planes, with separate lines shown for $v_{min}$, $v_{max}$ and $w_{VRH}$. Crystallographic axes <a>, <b>, and <c>
shown in pink. VRH = Voigt-Reuss-Hill average value of $v$.

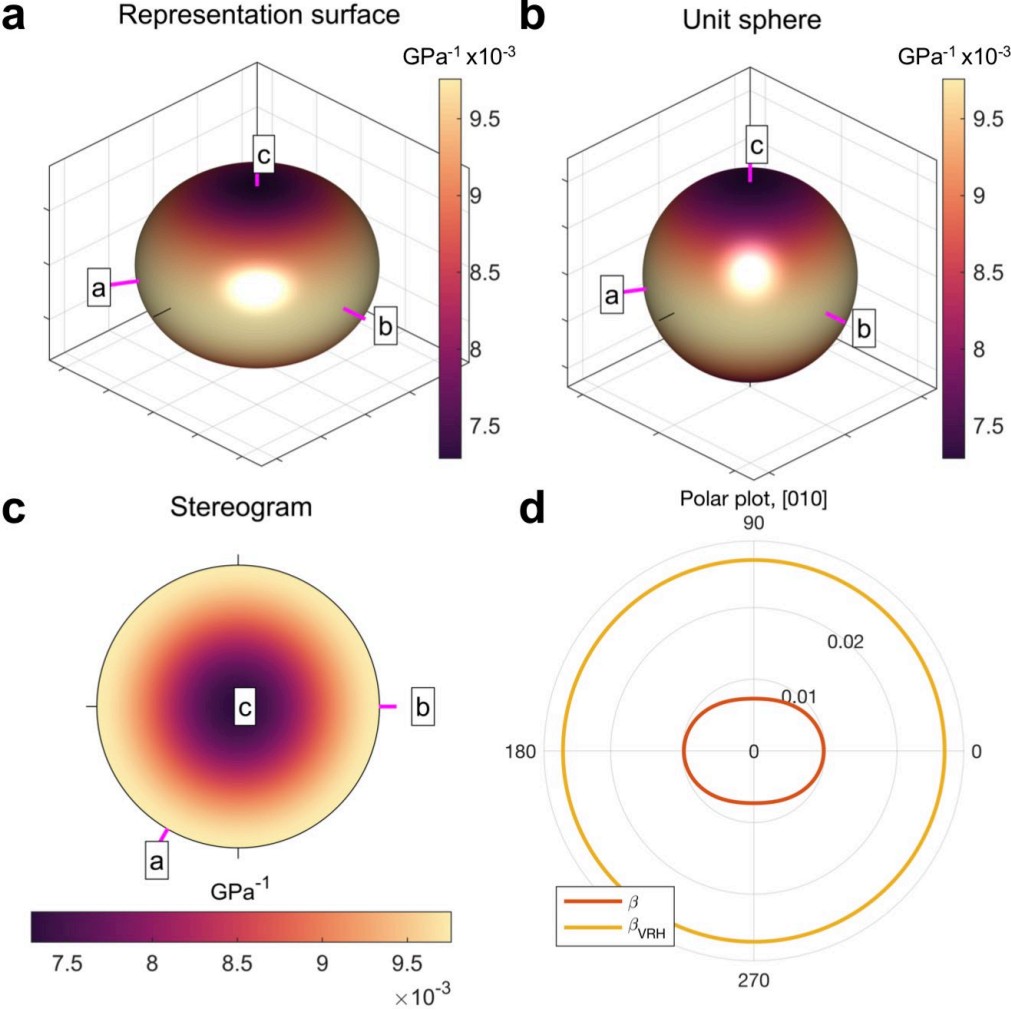


**Figure 7.** Alternative visualisations of the anisotropy of linear compressibility ($\beta$, in GPa$^{-1}$) of α-
quartz. **a**) 3D representation surface where the radius in any direction is proportional to the
magnitude of $\beta$. **b**) Projection of $\beta$ on to a unit sphere, colour coded by magnitude. **c**) Lower
hemisphere, equal area stereographic projection. **d**) Polar plot of anisotropy of $\beta$ in the [010] plane.
Crystallographic axes <a>, <b>, and <c> shown in pink. VRH = Voigt-Reuss-Hill average value of
$\beta$.
The linear compressibility ($\beta$) of an anisotropic crystal quantifies the directional response to an
applied hydrostatic load i.e. to pressure, not stress. For isotropic materials, the compressibility is a



scalar – directionally invariant – and is simply the inverse of the bulk modulus $K$ $(\beta = 1 / K)$.  For
anisotropic rock forming minerals, this is no longer the case and $\beta$ varies with direction.  Figure 7
shows the variation for $\alpha$-quartz using the same types of plots as for Young's modulus (Figure 5).
In summary, we note that as a corollary of the point made by Nye (1985) that no single surface can
represent the full richness of the 4[th] rank elasticity tensor, neither can any one measure (e.g. $E$, $G$, $\nu$
or $\beta$) convey the complete behavior of an anisotropic mineral.  The anisotropies of the different
parameters (through these plots) should be used in combination to understand a specific problem.
*Visualising second-rank tensors: stress and strain*
To address the challenges in visualizing stress and strain described above, we use two separate
graphical depictions, or glyphs, for the normal and shear components of the strain and stress tensors
(Kratz et al., 2014).  We use the Reynolds glyph for normal strains and stresses, as this can show
positive and negative principal values (Moore et al., 1996).  We use the HWY glyph to visualise the
shear components of the strain and stress tensors (Hashash et al., 2003).  Figures 8 and 9 show
examples of the Reynolds and HWY glyphs for strains and stresses, respectively.  Isotropic
compaction plots as a single point in Mohr space (Figure 8a), and as a sphere using a Reynolds
glyph (Figure 8b; shear strains are zero and so there is no HWY glyph).  For a general triaxial strain
with both shortening and stretching components, the Reynolds and HWY glyphs are shown in
Figure 8d and 8e. Note that in the HWY glyph for shear strain the maxima are located at 45° to the
principal axes, and the minima (0) are located along the principal axes.  Triaxially compressive
stress is shown in Figure 9a-c.  Again, maxima of shear stress in the HWY glyph are at 45° to the
directions of the principal (normal) stresses.  For a general triaxial stress with components of
compression and tension, the directional variations of normal and shear stress are shown in Figure
9d-f.




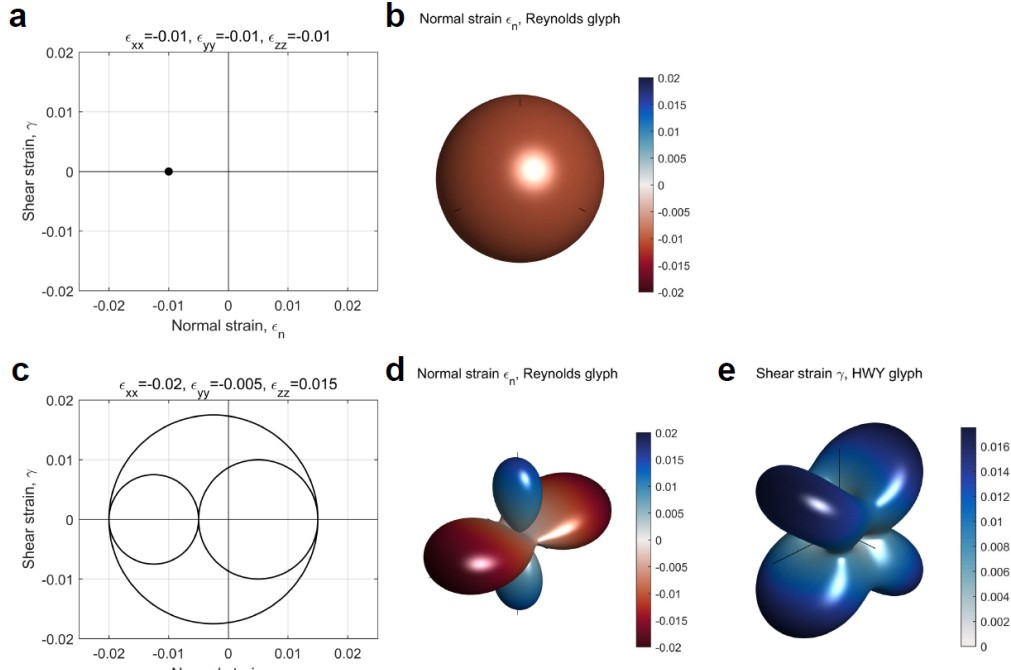


**Figure 8.** Examples of strain tensors depicted in Mohr space ($\varepsilon_n$, $\gamma$), and as Reynolds (normal
strains, $\varepsilon_n$) and HWY (shear strains, $\gamma$) glyphs. **a-b**) Isotropic compaction (taken as negative, blue
colour). **c-e**) Visualisations for a general triaxial strain. Note the lobes of extensional (blue) and
contractional (red) strain in the normal strain plot (**d**).



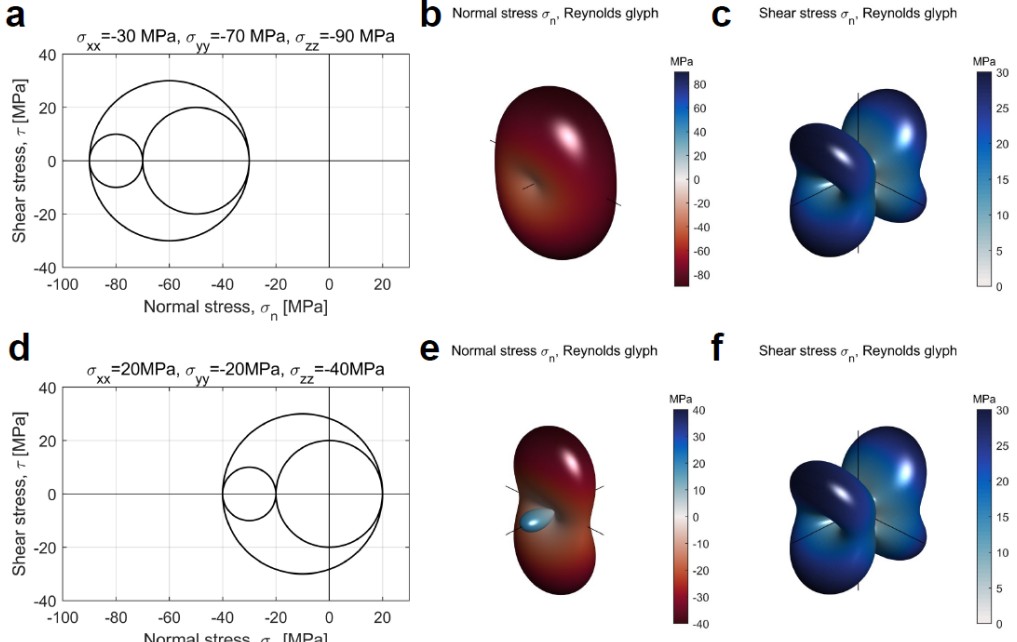

**Figure 9.** Examples of stress tensors depicted in Mohr space ($\sigma_n$, $\tau$) and as Reynolds (normal stress, $\sigma_n$) and HWY (shear stress, $\tau$) glyphs. **a-c**) Triaxial compression (taken as negative, blue colour). **d-f**) General triaxial stress with one principal stress tensile ($\sigma_{xx}$).

*Data sources*

The elastic properties of the minerals used in this study have been derived from previous compilations and original sources where possible. Many compilations of elastic and other physical properties are now available: see Bass (1995) and Almqvist & Mainprice (2017), and references therein. Note that most elastic properties are measured by laboratory methods whereas a minority are calculated from theory (*ab initio*). Single mineral lattice parameters have been extracted from the same publication as the elasticity data where possible, but if this was not available, we took representative values from Deer, Howie & Zussman (1992).

**4. Results – General trends**

From our database of published elastic properties of rock-forming minerals (246 data files covering 86 distinct minerals, all included with AnisoVis), we have calculated the maxima and minima for Young's modulus, Poisson's ratio, shear modulus and linear compressibility. In Figure 10 we show the variation in the anisotropy of Young's modulus ($E$) for 246 rock forming minerals as a function of $A^U$. If we consider a simple measure of the anisotropy of $E$ as the ratio between the maximum and minimum values, it is clear that most minerals display significant anisotropy with $E_{max}/E_{min}$ often greater than 2. With increasing $A^U$, many minerals show $E_{max}/E_{min}$ ratios of about 4. Figure 11 shows the anisotropy of shear modulus ($G$) for the same rock forming minerals, plotted against





$A^U$. The anisotropy of $G$, simply defined as $G_{max}/G_{min}$, is less than that shown for $E$, and there is a
general pattern of decreasing anisotropy of $G$ with increasing $A^U$.

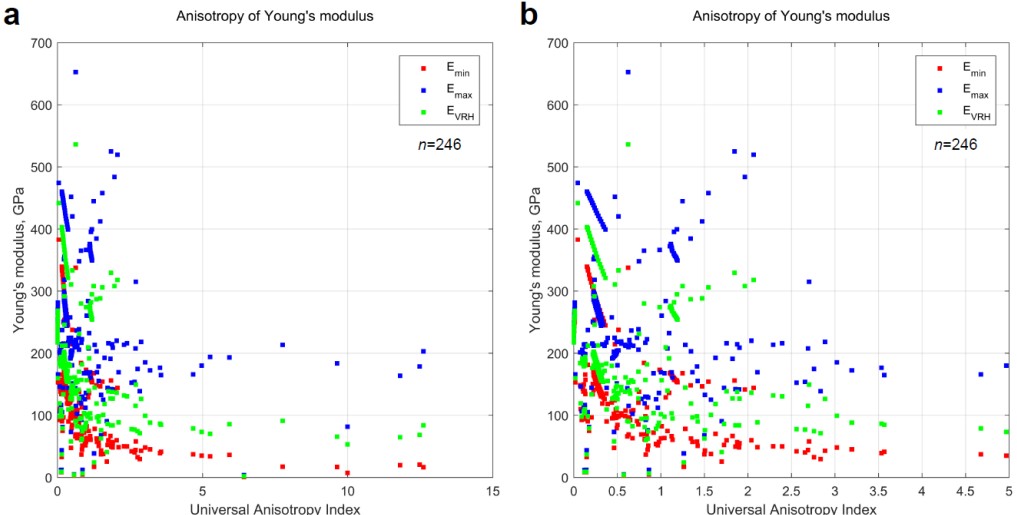

**Figure 10.** Anisotropy of Young's modulus in rock-forming minerals ($n$=246) plotted against the
Universal Anisotropy Index ($A^U$) of Ranganathan & Ostoja-Starzewski (2008). $E_{VRH}$ is the Voigt-
Reuss-Hill average of $E$. Many minerals display anisotropy of $E$ ($E_{max}/E_{min}$) of 2 or more. **b**) Close-
up of data in **a**) for UAI up to 5.

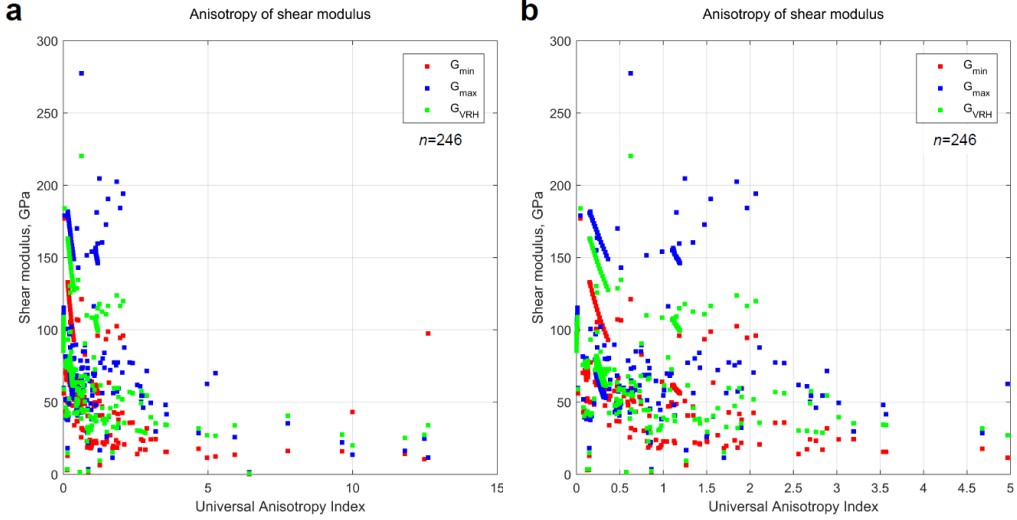

**Figure 11.** Anisotropy of shear modulus in rock-forming minerals ($n$=246) plotted against the
Universal Anisotropy Index of Ranganathan & Ostoja-Starzewski (2008). $G_{VRH}$ is the Voigt-Reuss-
Hill average of $G$. **b**) Close up of data in **a**) for UAI up to 5.



Figure 12 shows the variation in Poisson's ratio ($\nu$) versus $A^U$ for all minerals. The shaded area in
Figure 12a and 12b denotes the range $0 \leq \nu \leq 0.5$. As noted by Ting & Chen (2005), $\nu$ for
anisotropic materials can have no bounds. The data show that many minerals have minimum values
less than 0 and maximum values greater than 0.5. The histogram in Figure 13 shows the statistical
variation in $\nu_{min}$ for all minerals: 28% (=70/246) have negative minimum values for Poisson's ratio
– that is, they display auxetic behaviour. Analysis of the variation of $\nu_{max}$ shows that 37%
(=91/246) have values greater than 0.5 (Figure 13b). The mean value of the Voigt-Reuss-Hill
average of Poisson's ratio for all minerals is 0.2464 (Figure 13c), close to the default assumption of
many simplifications to elastic isotropy ($\nu$=0.25). A full list of the rock forming minerals in our
database that show auxetic behaviour is shown in Table 2, and the specific directions of negative $\nu$
are shown for several examples in the stereograms in Figure 14.

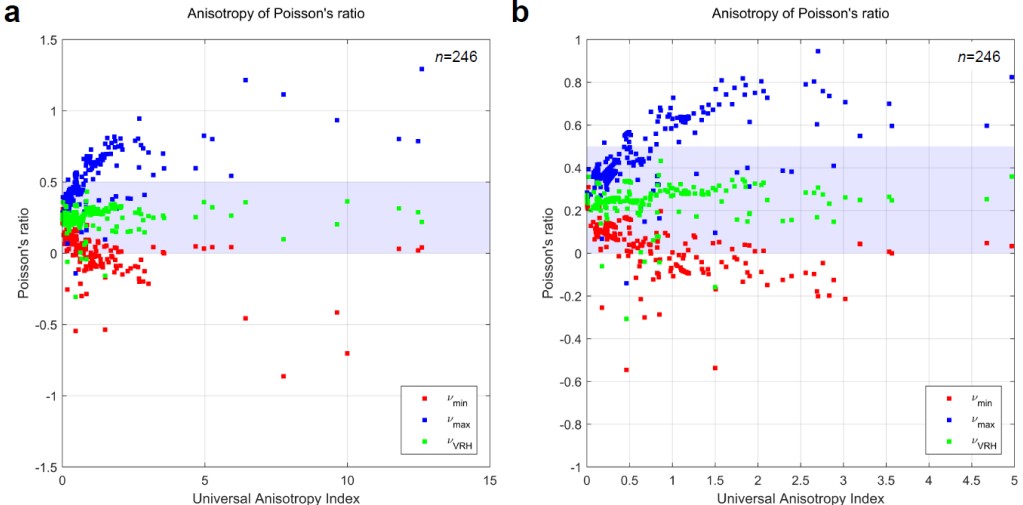

**Figure 12. a)** Anisotropy of Poisson's ratio in rock-forming minerals (n=246) plotted against the
Universal Anisotropy Index of Ranganathan & Ostoja-Starzewski (2008). $\nu_{VRH}$ is the Voigt-Reuss-
Hill average of $\nu$. **b)** Close up of data in **a)** for UAI up to 5.

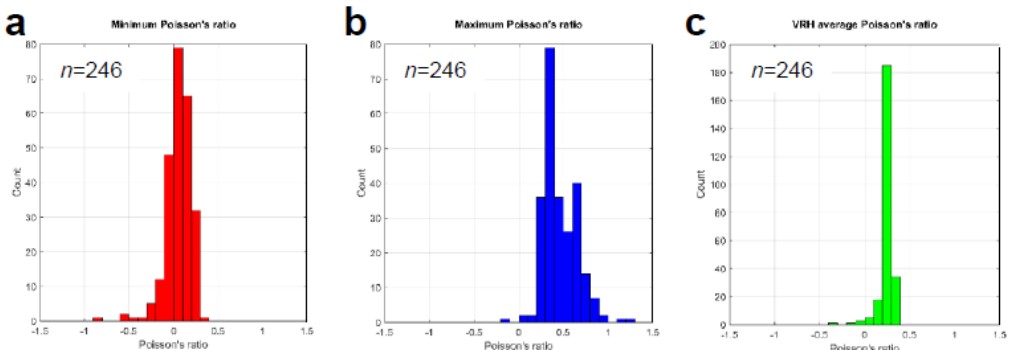

**Figure 13. a)** Histogram of $\nu_{min}$ values shown in Figure 12. Note that 28% (n=70/246) of minerals





display negative $\nu_{min}$. **b)** Histogram of $\nu_{max}$ values. 37% (n=91/246) minerals display $\nu_{max} > 0.5$. **c)**
Histogram of $\nu_{VRH}$ values. Mean $\nu_{VRH} = 0.2464$, very close to the common default assumption of $\nu$
$= 0.25$.

| Mineral | Symmetry | Minimum $\nu < 0$ | Minimum areal $\nu < 0$ | Reference |
|---|---|---|---|---|
| Albite (An0) | Triclinic | -0.03 | | Hearmon, 1984 |
| | Triclinic | -0.15 | | Brown et al., 2016 |
| Anhydrite | Orthorhombic | -0.046 | | Hearmon, 1979 |
| Andesine (An37) | Triclinic | -0.091 | | Brown et al., 2016 |
| Andesine (An48) | Triclinic | -0.075 | | Brown et al., 2016 |
| Antigorite | Monoclinic | -0.215 | | Bezacier et al., 2010 |
| Aragonite | Orthorhombic | -0.061 | | Hearmon, 1979 |
| Augite | Monoclinic | -0.012 | | Alexandrov et al., 1964 |
| Bytownite (An78) | Triclinic | -0.053 | | Brown et al., 2016 |
| Calcite | Trigonal | -0.047 | | Babuska & Cara, 1991 |
| | Hexagonal | -0.02 | | Chen et al., 2001 |
| Coesite | Monoclinic | -0.108 | | Weidner & Carleton, 1977 |
| α-Cristobalite | Tetragonal | -0.537 | -0.262 | Pabst & Gregorova, 2013 |
| β-Cristobalite | Cubic | -0.288 | -0.162 | Pabst & Gregorova, 2013 |
| Dolomite | Trigonal | -0.064 | | Hearmon, 1979 |
| Hornblende | Monoclinic | -0.075 | | Hearmon, 1984 |
| Illite-Smectite | Monoclinic | -0.416 | | Militzer et al., 2011 |
| Labradorite | Triclinic | -0.085 | | Ryzhova, 1964 |
| Labradorite (An60) | Triclinic | -0.009 | | Brown et al., 2016 |
| Labradorite (An67) | Triclinic | -0.025 | | Brown et al., 2016 |
| Lawsonite | Orthorhombic | -0.088 | | Sinogeikin et al., 2000 |
| Microcline | Triclinic | -0.199 | -0.042 | Babuska & Cara, 1991 |
| Oligoclase (An25) | Triclinic | -0.098 | | Brown et al., 2016 |
| Orthoclase | Monoclinic | -0.169 | | Hearmon, 1984 |
| | Monoclinic | -0.092 | | Waeselmann et al., 2016 |
| α-Quartz | Trigonal | -0.97 | -0.071 | Ogi et al., 2006 |
| | Trigonal | -0.93 | -0.067 | Babuska & Cara, 1991 |
| T=200°C | Trigonal | -0.123 | -0.088 | Lakshtanov et al., 2007 |
| T=400°C | Trigonal | -0.215 | -0.138 | Lakshtanov et al., 2007 |
| T=500°C | Trigonal | -0.301 | -0.186 | Lakshtanov et al., 2007 |
| T=573°C | Trigonal | -0.546 | -0.398 | Lakshtanov et al., 2007 |
| T=575°C | Hexagonal | -0.255 | -0.095 | Lakshtanov et al., 2007 |
| Rutile | Tetragonal | -0.044 | | Manghnani, 1969 |
| Sanidine | Monoclinic | -0.097 | | Waeselmann et al., 2016 |
| Sillimanite | Orthorhombic | -0.001 | | Verma, 1960 |
| Sphalerite | Cubic | -0.025 | | Hearmon, 1984 |
| Spinel | Cubic | -0.07 | | Hearmon, 1984 |





| | | | | |
|---|---|---|---|---|
| T=300°K | Cubic | -0.081 | | Anderson & Isaak, 1995 |
| T=350°K | Cubic | -0.079 | | Anderson & Isaak, 1995 |
| T=400°K | Cubic | -0.083 | | Anderson & Isaak, 1995 |
| T=450°K | Cubic | -0.083 | | Anderson & Isaak, 1995 |
| T=500°K | Cubic | -0.084 | | Anderson & Isaak, 1995 |
| T=550°K | Cubic | -0.084 | | Anderson & Isaak, 1995 |
| T=600°K | Cubic | -0.085 | | Anderson & Isaak, 1995 |
| T=650°K | Cubic | -0.033 | | Anderson & Isaak, 1995 |
| T=700°K | Cubic | -0.088 | | Anderson & Isaak, 1995 |
| T=750°K | Cubic | -0.089 | | Anderson & Isaak, 1995 |
| T=800°K | Cubic | -0.09 | | Anderson & Isaak, 1995 |
| T=850°K | Cubic | -0.092 | | Anderson & Isaak, 1995 |
| T=900°K | Cubic | -0.093 | | Anderson & Isaak, 1995 |
| T=950°K | Cubic | -0.094 | | Anderson & Isaak, 1995 |
| T=1000°K | Cubic | -0.095 | | Anderson & Isaak, 1995 |
| Staurolite | Orthorhombic | -0.201 | | Hearmon, 1979 |
| Stishovite | Tetragonal | -0.04 | | Babuska & Cara, 1991 |
| Talc (c1) | Triclinic | -0.864 | -0.287 | Mainprice et al., 2008 |
| P=0.87 GPa | Triclinic | -0.178 | -0.001 | Mainprice et al., 2008 |
| P=1.96 GPa | Triclinic | -0.107 | | Mainprice et al., 2008 |
| P=3.89 GPa | Triclinic | -0.009 | | Mainprice et al., 2008 |
| Talc (c2c) | Monoclinic | -0.126 | -0.029 | Mainprice et al., 2008 |
| P=0.15 GPa | Monoclinic | -0.107 | -0.021 | Mainprice et al., 2008 |
| P=0.35 GPa | Monoclinic | -0.125 | -0.025 | Mainprice et al., 2008 |
| P=0.64 GPa | Monoclinic | -0.091 | -0.002 | Mainprice et al., 2008 |
| P=0.93 GPa | Monoclinic | -0.028 | | Mainprice et al., 2008 |
| P=1.72 GPa | Monoclinic | -0.019 | | Mainprice et al., 2008 |
| Zircon (metamict) | Tetragonal | -0.113 | | Hearmon, 1984 |
| Zoisite | Orthorhombic | -0.014 | | Mao et al., 2007 |
| | | | | |
| Number of distinct minerals | | *n*=33 | *n*=7 | |


**Table 2.** List of rock forming minerals showing auxetic behaviour (Poisson's ratio < 0) in at least
one direction. Also shown are those minerals with directions that have negative areal Poisson's
ratio (Guo & Wheeler, 2006). The Reference column shows the source of the elasticity data for
each mineral used in the calculation. The auxetic directions were found by calculating Poisson's
ratio for every possible direction ($\alpha$, $\beta$, $\theta$ in the Turley & Sines reference frame shown in Figure 1)
using an angular increment of 1 degree in each direction.


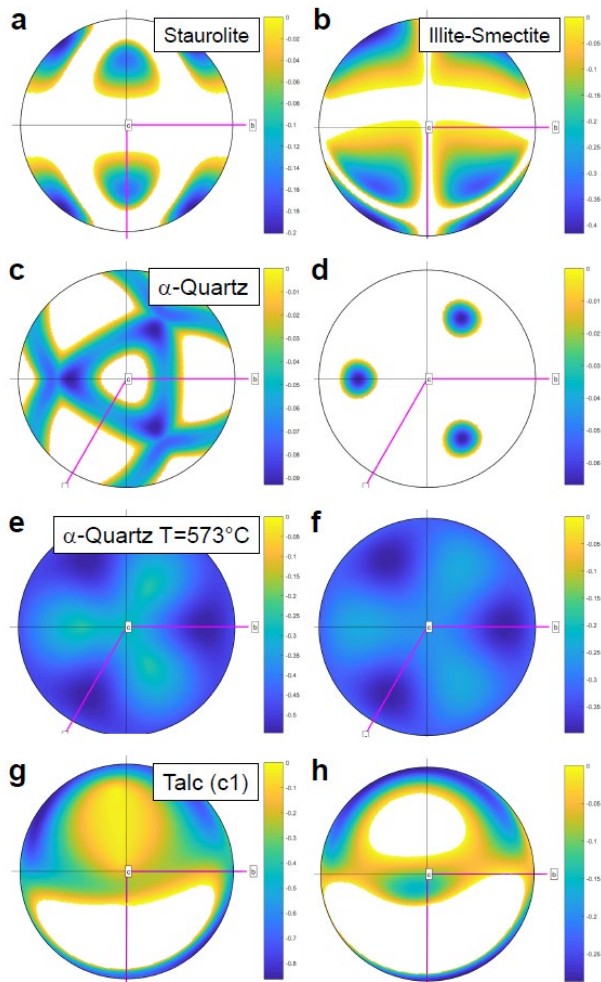


**Figure 14.** Examples of rock forming minerals showing auxetic and areally auxetic behaviour. Stereograms are all lower hemisphere, equal area projections and only the directions with negative Poisson's ratio (a, b, c, e, g) or negative areal Poisson's ratio (d, f, h) are shown coloured in (i.e. other directions show positive values). Crystallographic axes in pink. **a)** Staurolite. **b)** Illite-smectite. **c-d)** $\alpha$-Quartz. **e-f)** $\alpha$-Quartz at the temperature of the phase transformation to $\beta$-Quartz (hexagonal). **g-h)** Talc (c1, triclinic).


The elastic properties of minerals are known to be temperature ($T$) and pressure ($P$) dependent.
However, systematic data to quantify the variation of anisotropic elasticity with $T$ or $P$ is relatively
scarce. We summarise some of the published data in Figure 15, shown as the calculated range in
Poisson's ratio ($v_{min}$ to $v_{max}$). In terms of pressure dependence, the effect of increasing $P$ is to
decrease the anisotropy in $v$ for talc to within the range normally expected for isotropic minerals.
The opposite effect is observed for zircon, with modest increases in $v_{max}$ with $P$. The temperature
dependence of elastic anisotropy in quartz is well known (Mainprice & Casey, 1990), with a



significant excursion into auxetic behaviour at the temperature of the $\alpha$-$\beta$ phase transition at 573°C
(846°K).  The effect of increasing $T$ on the anisotropy of $\nu$ for olivine, corundum and spinel is
almost non-existent.

**Figure 15.** Anisotropy of Poisson's ratio in rock-forming minerals as a function of P (top) and T
(bottom).  Other than the well-known auxeticity of $\alpha$-$\beta$ quartz around the phase transition



(T=573°C), most minerals display Poisson's ratios of between 0-0.5. Talc (c1, triclinic) is one
exception, and the anisotropy of Poisson's ratio decreases markedly with increasing P.
Linear compressibility ($\beta$) also displays significant anisotropy in rock forming minerals (Figure 17).
A list of the rock forming minerals in our database that show negative linear compressibility (NLC)
is shown in Table 3. These minerals have directions that expand in response to a compressive
hydrostatic pressure (and vice versa: 'stretch-densification' of Baughman et al., 1998b). The
specific directions of negative $\beta$ are shown in the stereograms in Figure 16.

| Mineral | Symmetry | Minimum $\beta < 0$, GPa$^{-1}$ | Reference |
|---------|----------|-------------------------------|-----------|
| Lizardite | Hexagonal | -0.00165 | Reynard et al., 2007 |
| Talc (c1) | Triclinic | -0.00251 | Mainprice et al., 2008 |

**Table 3.** List of rock forming minerals showing negative linear compressibility (NLC) in at least
one direction.

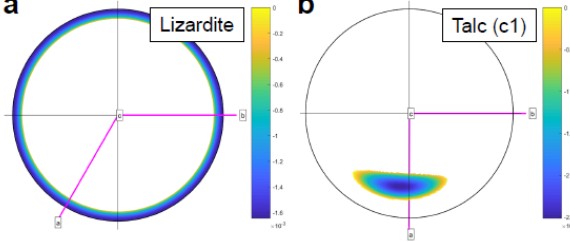


**Figure 16.** Rock forming minerals showing negative linear compressibility (NLC) in certain
directions. Stereograms are all lower hemisphere, equal area projections and only the directions
with NLC are shown coloured in (i.e. other directions show positive values). Crystallographic axes
in pink. **a**) Lizardite. **b**) Talc (c1, triclinic).

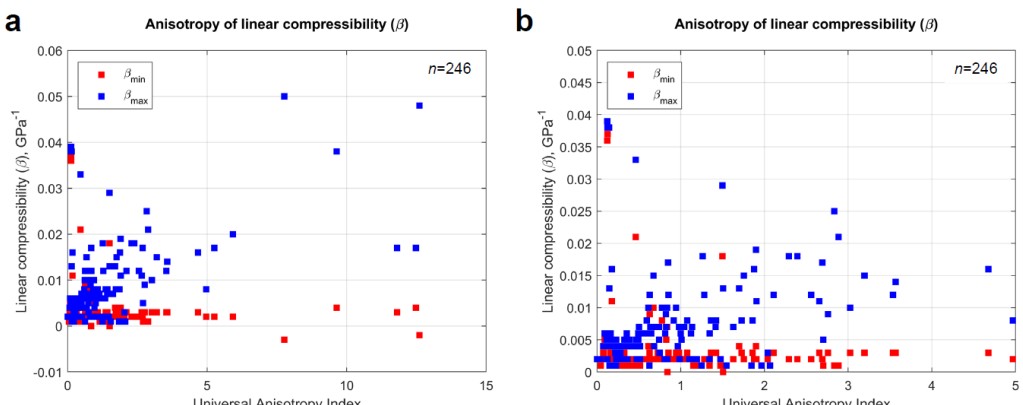

**Figure 17.** Anisotropy of linear compressibility in rock-forming minerals.





We can summarise the elastic anisotropy data for rock forming minerals using the Elastic
Anisotropy Diagram of Ranganathan & Ostoja-Starzewski (2008). In their review of Poisson's ratio
in materials, Greaves et al. (2011) used a plot of bulk modulus $K$ versus shear modulus $G$, however
for the anisotropic rock forming minerals there is no single value of either of these properties. We
therefore take the ratios KV/KR and GV/GR and plot these instead (Figure 18). Unsurprisingly,
minerals with monoclinic, triclinic and hexagonal symmetries dominate the higher anisotropies,
while minerals with cubic, orthorhombic and tetragonal symmetries are generally less anisotropic.

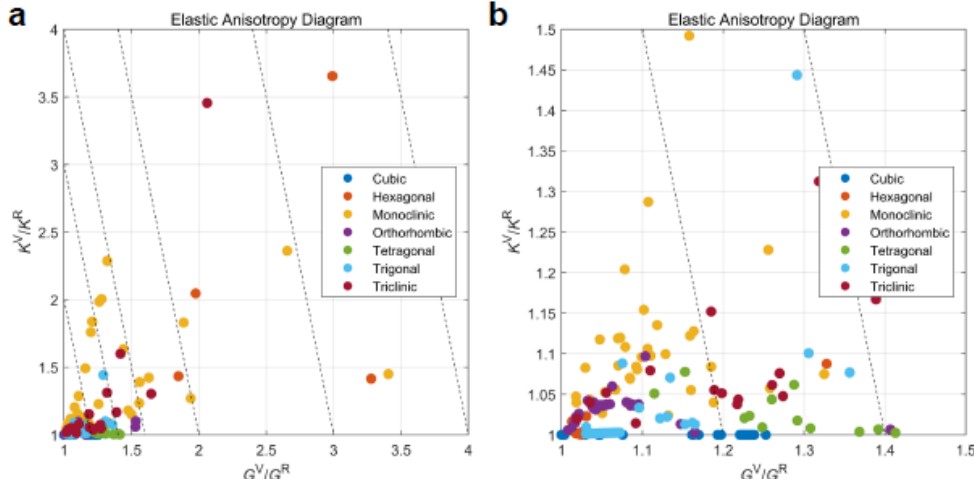

**Figure 18. a**) Anisotropy of rock-forming minerals (n=246) using the Elastic Anisotropy Diagram
used in materials science, grouped by mineral symmetry class. **b**) Close-up of the data plotted in a),
in the range $G^V/G^R$ 1 to 1.5 and $K^V/K^R$ 1 to 1.5.

**5. Results – Specific examples**
*Twinning*
Deformation or mechanical twinning critically depends on the anisotropy of elastic properties
because minerals respond elastically to imposed stress (or strain) before exceeding the threshold for
twin nucleation and propagation (Christian and Mahajan, 1995, and references therein). Perhaps the
most widely accepted theory is that twin initiation occurs when an applied shear stress along the
twin shear plane ($K_1$) in the shear direction of twinning ($\eta_1$) reaches a critical value (critically
resolved shear stress, CRSS) for twin nucleation and propagation, analogous to Schmid's law for
dislocation slip (Thompson and Millard, 1952; Bell and Cahn, 1953; Christian and Mahajan, 1995).
However, experimental results can indicate that twinning dynamics can be more complex (e.g., Bell
and Cahn, 1957). Additional complexities, such as energy barriers for the nucleation of coeval
defects such as stacking faults, disconnections, and unstable transition states associated with
twinning, have also been considered for twinning in metals (e.g., Serra & Bacon; 1996; Kibey et al.,
2007; Pond et al., 2016). Development of a general theory of mechanical twinning applicable to
most minerals is still lacking. Nevertheless, shear modulus $G$ in $\eta_1$ along $K_1$ is highly relevant to
mechanical twinning.





Dauphiné twins in $\alpha$-quartz are merohedral twins, meaning only some atoms exchange their
positions, resulting in a host-twin symmetry relationship that can be described simply by a 180°
rotation about the c-axis, and recognisable in EBSD maps via a 60° misorientation around the c-
axis. The formation of Dauphiné twins has been related to the difference in elastic strain energy
between twinned and un-twinned at constant stress (Thomas & Wooster, 1951; Tullis, 1970; De
Vore, 1970). This difference in elastic strain energy can be written as
$\qquad \Delta E = \frac{1}{2} (\sigma_1 - \sigma_3)^2 \, \Delta s_{11}'$ (15)
where $(\sigma_1 - \sigma_3)$ is the applied differential stress, and $\Delta s_{11}' = s_{11}'_{\text{twinned}} - s_{11}'_{\text{un-twinned}}$. Note that $s_{11}'$ is
the reciprocal of the Young's modulus for a given direction. Dauphiné twinning occurs more
readily in those directions for which the strain energy difference ($\Delta E$) is larger, under a boundary
condition of constant axial stress (the inverse is also true: under a condition of constant strain, the
preferred directions of twinning are those that minimise $\Delta E$ (Paterson, 1973)). The variation of
$\Delta s_{11}'$ with direction in $\alpha$-quartz is shown in Figure 19. The stereogram is the same pattern shown in
Thomas & Wooster (1951; their Figure 3a) and Tullis (1970; her Figure 2b). Also shown is a 3D
representation surface of $\Delta s_{11}'$, which emphasises the anisotropy of favoured directions for
Dauphiné twins in $\alpha$-quartz. The significance of Dauphiné twinning in quartz has recently been
described for sandstones compacted during diagenesis (Mørk and Moen, 2007), deformed in fault
damage zones (Olierook et al., 2014), and deformed by meteorite impact (Wenk et al., 2011; Timms
et al., 2019; Cox et al., 2019), and granitoid protomylonites (Menegon et al., 2011). In all cases,
Dauphiné twins can be used to infer palaeostresses from deformed microstructures. In addition,
Menegon et al. (2011) make the point that Dauphiné twins, formed early in a deformation history,
may effectively store strain energy which is then consumed in later plastic deformation
mechanisms. De Vore (1970) plotted the directional variation of compliances for quartz, ortho- and
clino-pyroxene, hornblende and plagioclase and thereby extended the initial concept of Thomas &
Wooster (1951). To our knowledge, detailed analyses of mechanical twins in these phases has not
yet been related to the anisotropy of elastic compliance or the calculated variations in elastic strain
energy for specific applied loads.

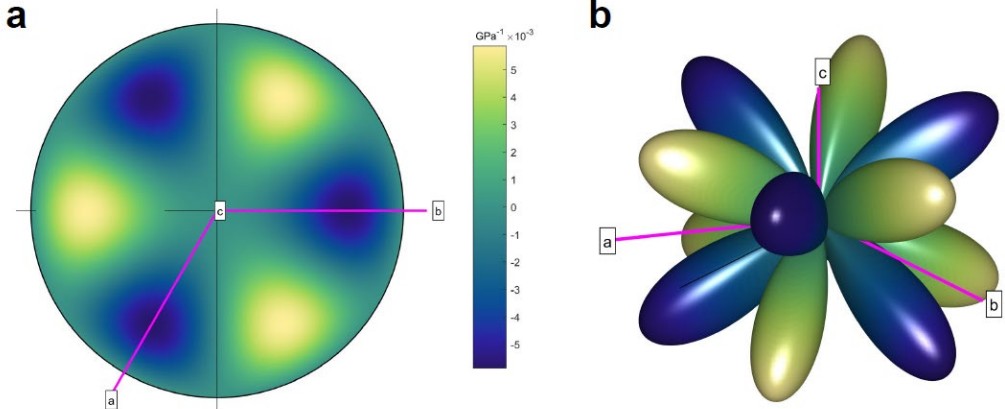

**Figure 19.** Anisotropy of $\Delta s_{11}'$ for Dauphiné twinning in $\alpha$-quartz. $\Delta s_{11}'$ is the difference in the
compliance $s_{11}'$ between the twinned and un-twinned orientations for each direction. **a)** Stereogram
(lower hemisphere, equal area projection) and **b)** a 3D representation surface, both with the



crystallographic reference axes marked. The directions represented by pale yellow/green colours
will be favoured for twinning, whereas the directions shown in blue will not.
The relationship between elastic anisotropy and deformation twinning has been investigated in
zircon (Timms et al., 2018). In zircon, deformation twins can form as a response to shock
conditions and are diagnostic of hypervelocity impact events (Timms et al., 2012; 2017; Erickson et
al. 2013). Shock twinning in zircon, which is tetragonal, can occur in up to four symmetrically
equivalent orientations, forming along {112} composition planes (the of invariant shear, or $K_1$), and
with shear direction $\eta_1$ = <111>, resulting in a host-twin 65° / {110} misorientation relationship
(Timms et al., 2018). Twinning in this mode has been shown to correspond to the lowest values of
$G$ ($G_{min}$ = $G_{<111>}$ = ~98 GPa) (Timms et al., 2018). Furthermore, the lowest values of $\nu$ are along
<111> in zircon, indicating that zircon is almost perfectly compressible in <111> ($\nu_{min}$ = $\nu_{<111>}$ >0
and <<0.1) (Timms et al., 2018). These authors illustrate that elastic softness in shear (low $G$) and a
lack of lateral strain in the shear plane ($\nu$ ~ 0) are favorable conditions for twinning in zircon
(Timms et al. 2018). However, further work is required to determine the critically-resolved shear
stress for twinning in zircon. Nevertheless, the ability to calculate and visualize anisotropic elastic
properties in specific crystallographic directions presented here will be very useful for detailed
investigations of mechanical twinning in other phases.
*Polymorphic phase transformations*
Coherent phase transformations (or transitions) may also be related to the anisotropy of elastic
properties, including the $\alpha$-$\beta$ transformation in quartz. Coe & Paterson (1969) describe experiments
on oriented cores from single crystals of quartz heated to temperatures above the transformation
temperature (573°C, at atmospheric pressure), and subjected to non-hydrostatic stress. They found
that the temperature of transition was raised by different amounts depending on the orientation of
the stress with respect to the crystal. Crystal cores stressed parallel to the $c$-axis showed the least
change, whereas those loaded in the $m$-direction (perpendicular to $c$) showed the greatest increase
(they also performed experiments on samples cored in the $o$ and $r'$ directions). The temperature of
phase transformation from $\alpha$- (trigonal) to $\beta$- (hexagonal) quartz is therefore stress dependent. The
theoretical analysis of Coe & Paterson (1969, their Appendix C) ascribes this dependence to an
infinitesimal reversible transformation strain, based on the formalism of Eshelby (1957, 1959).
Noting that the transformation is also marked by a '*dramatic increase in the development of small-*
*scale Dauphine twins*', we have calculated the elastic strain energy per unit volume for each of the
four core orientations tested by Coe & Paterson, using their values of applied stress ($\sigma_1$ = 1 GPa, $\sigma_2$
= $\sigma_3$ = 300 MPa; all compressive) and the elastic constants of $\alpha$-quartz at 500°C (Lakshtanov et al.,
2007). The results are shown in Figure 20, and clearly show an exact correlation with experimental
data: the sample loaded in the $m$ direction has the highest strain energy, and that in the $c$ direction
has the lowest. The overall sequence is $W(m) > W(r') > W(o) > W(c)$, which precisely mirrors that
of the variation in $\partial T/\partial \sigma$ listed for each direction in Coe & Paterson (1969, their Table 3).
Therefore, we speculate that the mechanism of phase transformation of $\alpha$- to $\beta$- quartz may be
similar to that of Dauphiné twinning in $\alpha$-quartz, and favoured for those directions that maximise
the elastic strain energy under a constant applied stress. We also note that similar processes may
occur in pyroxenes (Coe, 1970; Coe & Muller, 1973; Clement et al., 2018).


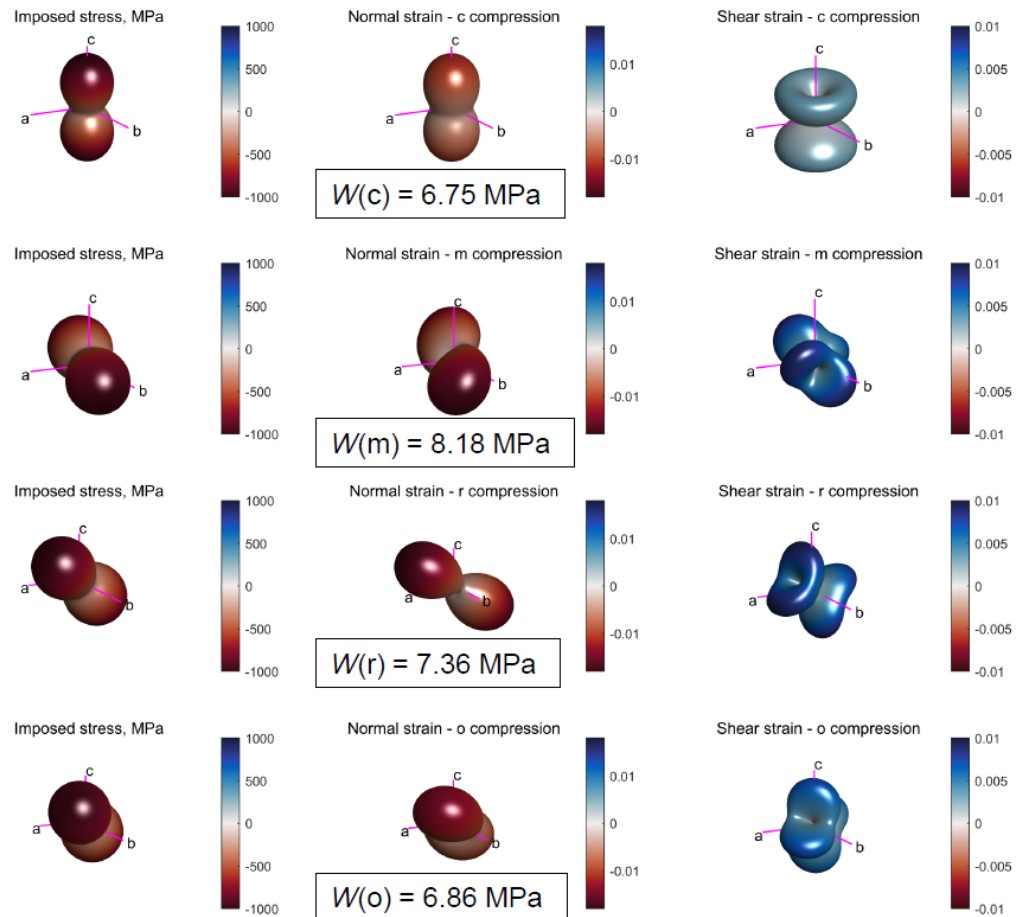

**Figure 20.** Variation in strain (normal and shear) and elastic strain energy for different applied
loads in α-quartz at 500 °C (Lakshtanov et al. 2007). The same compressive stress ($\sigma_1$ = -1000
MPa, $\sigma_2$ = $\sigma_3$ = -300 MPa) is applied along the $c$ (row 1), $m$ (row 2), $r$ (row 3), and $o$ (row 4)
directions in a single crystal. The Reynolds (2[nd] column) and HWY (3[rd] column) glyphs show the
normal and shear strains, respectively. The elastic strain energy per unit volume ($W$) is shown for
each configuration. Note that $W(m) > W(r) > W(o) > W(c)$.
Visualisation of elastic anisotropy has been used to gain new insights into the effects of intrinsic
elastic stiffness on the transformation from zircon to the high pressure $ZrSiO_4$ polymorph reidite
(Timms et al., 2018). The occurrence of lamellar reidite in shocked zircon from hypervelocity
impact structures has been observed to be spatially limited to low-U domains that have not
accumulated radiation damage of the lattice from the decay of U to Pb – a process known as
metamictization (Cavosie et al., 2015; Erickson et al., 2017). Using elastic constants measured for
variably metamict zircon (Özkan, 1976; Özkan and Jamieson, 1978), Timms et al. (2018) illustrated
that the process of metamictization significantly reduces maxima of $E$, $G$ and $\nu$ in zircon resulting
in a compliant, isotropic structure. These authors argued that metamict domains in zircon grains are
not elastically stiff enough to support sufficiently high stresses and pressures to facilitate the
transformation to reidite, limiting reidite lamellae to highly crystalline non-metamict domains



during the same shock event. This finding illustrates the dependance of elastic properties on lattice
defects and a potential role of intrinsic elastic properties in phase transofrmations.
*Metamorphic reactions and equilibrium thermodynamics*
The role of elastic deformation in the thermodynamics of preferred orientations and reactions at the
scale of individual grains has long been controversial (Macdonald, 1960; Brace, 1960; Kamb, 1961
and discussion thereof; Paterson, 1973; Wheeler, 2017). Debate has centred on the role, if any, of
the elastic strain energy, W. Macdonald (1960) and Brace (1960) defined the Gibbs free energy of
non-hydrostatically stressed minerals in terms of the elastic strain energy, and thereby implicitly
defined equilibrium under these conditions. They went on to assert that preferred orientations
would develop by the (re-)orientation of a crystals in a given stress system such that their elastic
strain energies were maximised. Wheeler (2017), following Kamb (1961) and Paterson (1973),
asserts that there is no definable equilibrium in non-hydrostatically stressed systems. Therefore, it
is wrong to equate the Gibbs energy for stressed systems of polycrystals to the elastic strain energy.
Moreover, the contribution of the elastic strain energy to the chemical potentials along stressed
interfaces, through the Helmholtz free energy term, is second order and therefore negligible
(Wheeler, 2018).
*Brittle cracking, decrepitation and dehydration*
The magnitude of stresses around fluid-filled pores and cracks developed within single crystalline
grains under load can be important for a variety of natural processes. The decrepitation of fluid
inclusions occurs when the stresses around the pore exceed the local tensile strength, and the fluid
will then drain away. Previous analyses have been rooted in linear elastic fracture mechanics, under
an assumption of elastic isotropy. Similarly, in reacting systems the dehydration of hydrous phases
can lead to pore fluid overpressures which crack the reacting grain and produce dehydration
embrittlement (e.g. Raleigh & Paterson, 1965; Jung et al., 2004). Accurate predictions of the stress
levels sustainable by intracrystalline pores and cracks are therefore vital to understanding these
fundamental mechanisms. Jaeger & Cook (1969; and repeated by Pollard & Fletcher (2005))
asserted that the elastic anisotropy of rocks, measured as the ratio of Young's moduli $E_{max}/E_{min}$, is
rarely as high as 2, and therefore the effects of elastic anisotropy are minor to negligible. Davis et
al. (2017) used 3D boundary element models to show that Poisson's ratio and void (pore or crack)
shape can exert significant control on the local stresses at the void-matrix boundary as a precursor
to tensile or shear failure.
We have calculated the circumferential stresses around crack-like voids developed within single
elastically anisotropic grains of selected minerals (Figures 21-22). The model configuration follows
that of Jaeger & Cook (1969; derived from Green & Taylor, 1939), and is based on a thin 2D
orthotropic plate with a single crack of aspect ratio 5:1. The assumption of orthotropy reduces the
required elastic constants to five ($E_1$, $E_2$, $G$, $\nu_{21}$, $\nu_{12}$). We calculated the appropriate values of $E$, $G$
and $\nu$ from polar plots of anisotropy for the [010] crystallographic plane in each mineral using
AnisoVis (see Figure 4d, 5d, 5e-f). For an applied uniaxial tensile load ($\sigma_0$ in Figure 21) and a
plane strain assumption, the resulting anisotropy of circumferential stress ($\sigma_{\theta\theta}$) at the void-matrix
boundary is shown for four different minerals in Figure 22. Each polar plot shows the $\sigma_{\theta\theta}$
normalised by the applied load $\sigma_0$ in the [010] plane, and for two different configurations of the
anisotropy with respect to the load: $\sigma_0$ parallel to the direction of $E_{max}$ (red curves), and





perpendicular to the direction of $E_{max}$ (blue curves). For both of the hydrous sheet silicates talc (c1;
Mainprice et al., 2008) and lizardite (Reynard et al., 2007), the stresses display significant
anisotropy (Figure 22a and b), with amplifications of 6-7 times the stress predicted by assuming the
crystal is isotropic (black curves, calculated with VRH averages of $E$ and $v$). These stresses are
likely significant for the failure of cracks or narrow fluid-filled pores in dehydrating subducting
slabs (Healy et al., 2009; Ji et al., 2018). For the two feldspar examples, albite (Brown et al., 2016)
and sanidine (Waeselmann et al., 2016), the amplification of circumferential stress is also
significant, at 4-5 times the isotropic prediction. Again, these stresses imply that fluid-filled pores
in phenocrysts of these phases may fail sooner than currently predicted under the assumption of
elastic isotropy. The restriction to 2D may appear limiting in these simple illustrative models, but
pending the development and analysis of fully 3D finite or boundary element models of stresses
around voids in elastically anisotropic media, they can provide useful insights into the relative
magnitude of local stresses and brittle failure. Moreover, we refute the suggestion from Jaeger &
Cook (1969) that as the anisotropy of Young's modulus in rocks is low, the anisotropy of stresses
around pores and cracks is therefore unimportant.

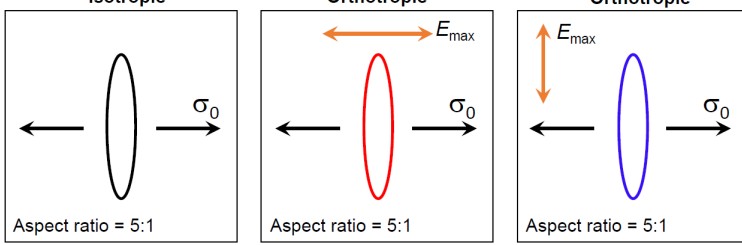


**Figure 21.** Schematic outline for models of narrow cracks in thin 2D orthotropic plates. The crack
is subjected to a uniaxial tensile stress, and plane strain is assumed. The colours of the crack
outlines correspond to the circumferential stress predictions in Figure 22.





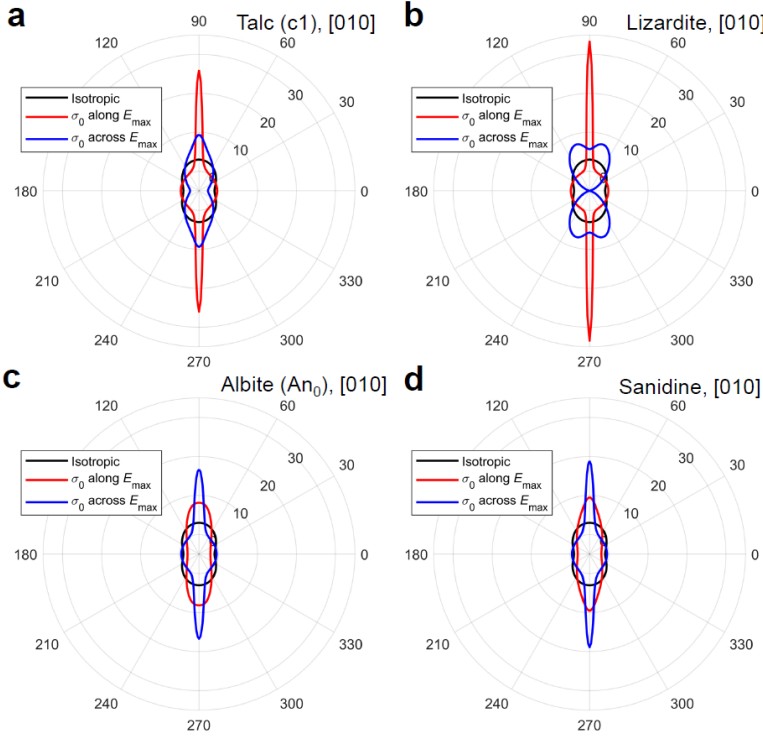


**Figure 22.** Predictions of stresses around cracks in thin 2D orthotropic plates. Curves show the
directional variations in the circumferential stress ($\sigma_{\theta\theta}$) normalised by the applied uniaxial tensile
load ($\sigma_0$). **a)** Talc (c1, triclinic), [010] plane. **b)** Lizardite, [010] plane. **c)** Albite, [010]. **d)**
Sanidine, [010].

**6. Summary**
We reiterate a key point made by Marmier et al. (2010) in their analysis of chemical compounds:
it's only by visualising elastic anisotropies, preferably in 3D, that we can truly perceive them and
quantify their directions; this then allows us to relate these elastic properties to the underlying
crystal structure and explore the consequences for their behaviour. In developing AnisoVis and
using it to quantify the anisotropy of a specific mineral, we have presented multiple alternative
visualisations of the directional variation of commonly used elastic properties such as Young's
modulus ($E$), Poisson's ratio ($\nu$), shear modulus ($G$) and linear compressibility ($\beta$). Used in
combination, these depictions serve to increase our understanding of the relationships between the
anisotropy of elastic properties and the underlying crystal symmetry and structure. We note that the
existence of directions with negative Poisson's ratios and negative linear compressibilities in certain
minerals (previously unreported). A potentially important consequence of these findings is that
there must also be specific directions along which these properties – Poisson's ratio or linear
compressibility – are 0. These directions will form surfaces in 3D which represents the boundary
between a) domains of positive and negative Poisson's ratio (both 'regular' and areal), along which
a uniaxially applied load will produce no lateral strain; and b) domains of positive and negative





linear compressibility, along which an applied hydrostatic load will produce no shortening or
stretching. These surfaces and directions in rock forming minerals may yet lead to new discoveries
in the physical behaviour of natural systems and novel applications in materials science or
engineering (e.g. Wu et al., 2015).
Considering the results from the database of 246 sets of elastic properties, we observe that:
• significant elastic anisotropy of rock forming minerals is much **more common** than previously
reported e.g. many minerals – 33 of the 86 we analysed – have auxetic directions, and some are
areally auxetic;
• the elastic anisotropy of rock forming minerals is **wider** than previously reported, with widely
assumed 'natural limits' frequently exceeded e.g. Poisson's ratio for many minerals is either < 0
or > 0.5.
For specific minerals, we also observe that
• elastic anisotropy has consequences for intracrystalline stresses under applied strain (and vice
versa); the difference between an assumption of isotropy and using the full elastic anisotropy is
often of the order of tens of MPa (even for small strains) – i.e. likely to be significant for the
deformation around voids such as pores and cracks, especially in dehydrating or decrepitating
systems;
• elastic anisotropy is important for mechanical (deformation) twining, especially Dauphiné
twinning in quartz but probably in other minerals too;
• coherent phase transformations, such as the $\alpha$-$\beta$ transition in quartz, show a clear correlation
with the magnitude of elastic strain energy per unit volume and the stress dependence of the
transition temperature.
*Further work*
We are not currently limited by data; we need to process the elasticity data we have and use it to
improve our understanding of Earth processes. In theoretical terms, perhaps the biggest advance
would come from a solution to the Eshelby problem for an anisotropic inclusion in an anisotropic
host, for ellipsoids of general shape and orientation, for the points inside and outside the inclusion.
This problem is non-trivial but would be of direct relevance to the inclusion-host studies estimating
pressure histories, and for mechanical problems involving voids and cracks in anisotropic crystals,
including reacting systems. Numerical modelling studies of the deformation around voids and
cracks might usefully incorporate a wider range of values of $E$ and $\nu$. Visualisation of direction-
specific elastic properties will be useful for future investigations of the mechanics of twinning,
dislocations, and fractures in a wide range of minerals. Earthquake focal mechanisms are known to
depend on the elastic anisotropy of the source region (Vavrycuk, 2005), and better understanding of
the anisotropies in rock forming minerals is informing models of fabrics in subducting slabs (Li et
al., 2018) and interpretations of microseismicity from commercial hydraulic fracturing operations
(Jia et al., 2018). A practical assessment of the contribution of elastic strain energy to metamorphic
reactions might involve the systematic mapping of major element chemistry around specific
inclusions.
We believe that publicly available and easy-to-use software tools like AnisoVis may be useful in
teaching environments to guide understanding of the links between mineral properties (elastic,



acoustic, optical) and their underlying symmetry and lattice structure. Following Nye's original
text, other properties such as piezolectric and thermal conductivities, could also be added and
visualised (Tommasi, 2001; Mainprice et al., 2015). Our AnisoVis MATLAB source code and
sample elasticity files have been made available in open repositories so that other developers and
researchers will optimise and extend the functionality, and that "given enough eyeballs, all bugs are
shallow" (Raymond, 1999).

**Code & Data Availability**
AnisoVis, including MATLAB source code, a basic user guide and data files for mineral elasticity
from published sources, is freely available on:
• GitHub (https://github.com/DaveHealy-Aberdeen/AnisoVis) and
• Mathworks FileExchange (https://uk.mathworks.com/matlabcentral/fileexchange/73177-
anisovis).

**Author Contribution**
DH designed the software, and wrote the code to calculate the anisotropic elastic properties. NET
contributed most of the section on twinning. MAP contributed to the code, especially the
calculation of directional properties in Cartesian and crystallographic reference frames. All authors
contributed to the manuscript.

**Competing Interests**
The authors declare that they have no conflict of interest.

**Acknowledgements**
DH thanks John Wheeler (Liverpool) for discussion, and Ross Angel (Padua) for discussion and a
reprint. This paper is dedicated to the memory of John Frederick Nye (1923-2019) whose seminal
text book, first published in 1957 (Physical Properties of Crystals: Their Representation by Tensors
and Matrices; reprinted as Nye, 1985), has been a huge influence on the lead author. DH
acknowledges financial support from NERC (UK), grant NE/N003063/1.

**Appendix A – benchmarks to previously published anisotropic elastic properties**
The outputs from AnisoVis, and the calculations underlying them, have been benchmarked against
previously published examples, chiefly from chemistry and materials science literature. Figures
produced by AnisoVis are shown below, with one example per symmetry group, formatted to
mimic the plots in the original publication.



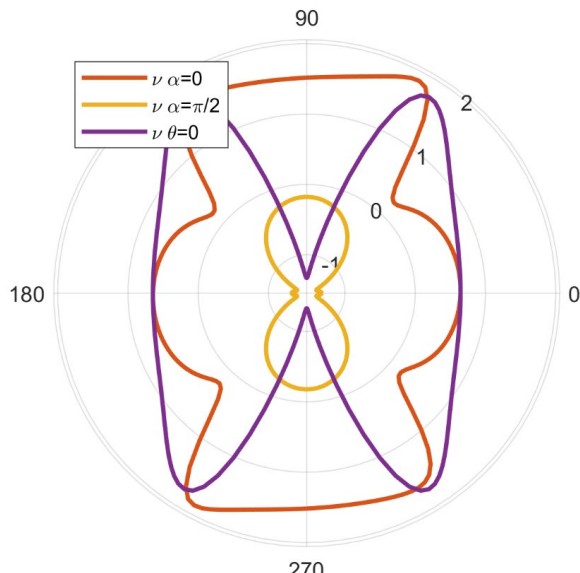


**Figure A1.** Benchmarks to Rovati (2004, their Figure 4) for monoclinic cesium dihydrogen
phosphate. Note the extreme auxeticity (negative Poisson's ratio) shown by this material.


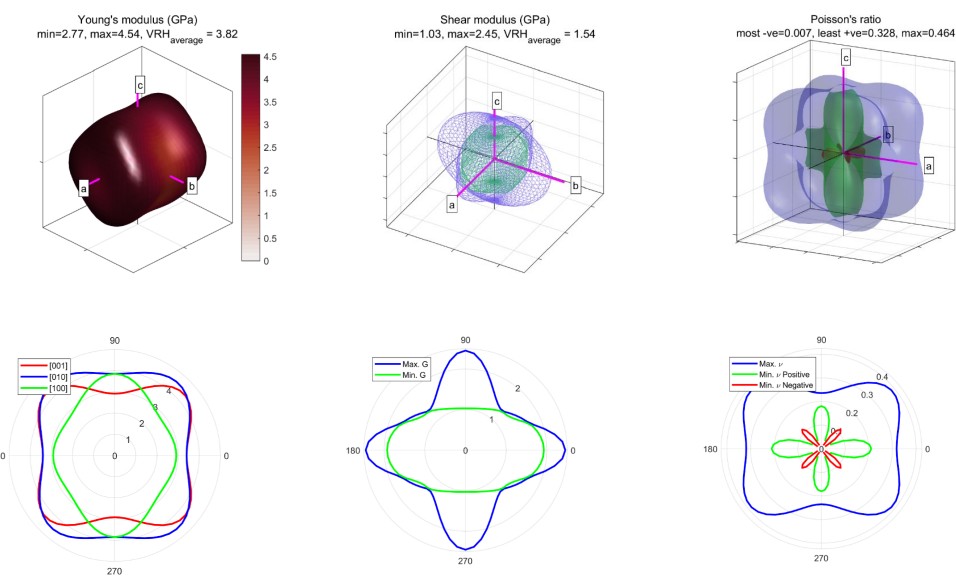

**Figure A2.** Benchmarks to Tan et al. (2015, their Figures 2, 3 and 4) for orthorhombic ZIF-4, a
zeolite. Plots shown for Young's modulus, shear modulus and Poisson's ratio.




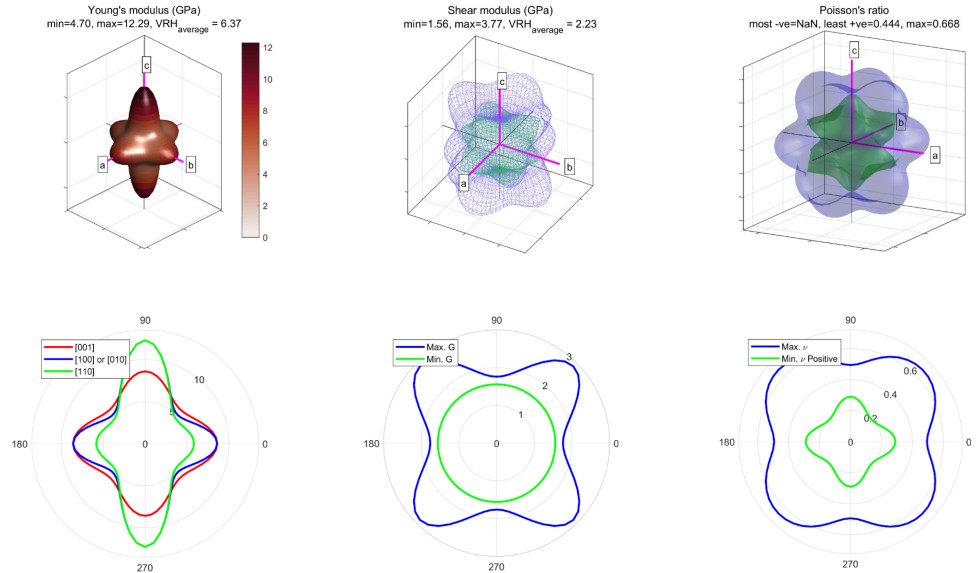

**Figure A3.** Benchmarks to Tan et al. (2015, their Figures 2, 3 and 4) for tetragonal ZIF-zni, a zeolite. Plots shown for Young's modulus, shear modulus and Poisson's ratio.

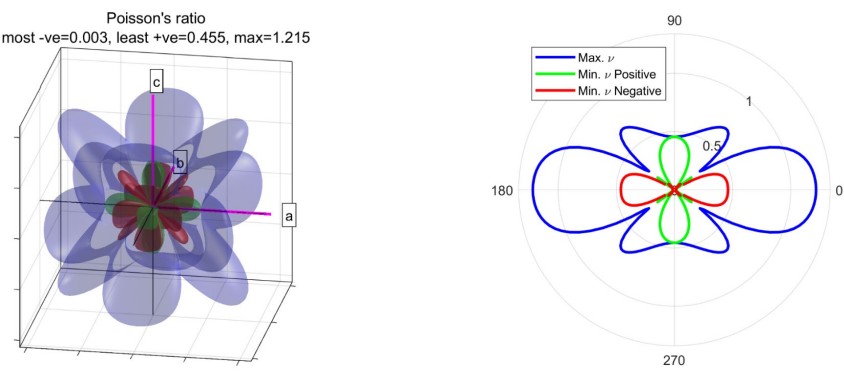

**Figure A4.** Benchmarks to Marmier et al. (2010, their Figure 5 and 6) for cubic cesium. Note the auxetic nature of Poisson's ratio.





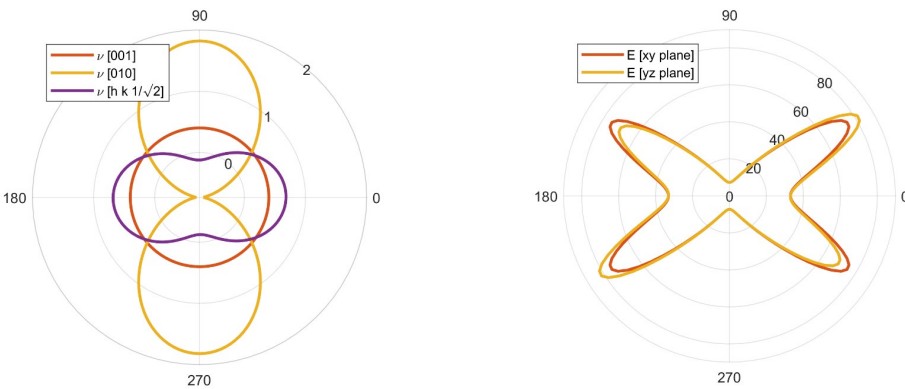

**Figure A5.** Benchmarks to Gunton & Saunders (1972, their Figures 3 and 6) for trigonal arsenic.

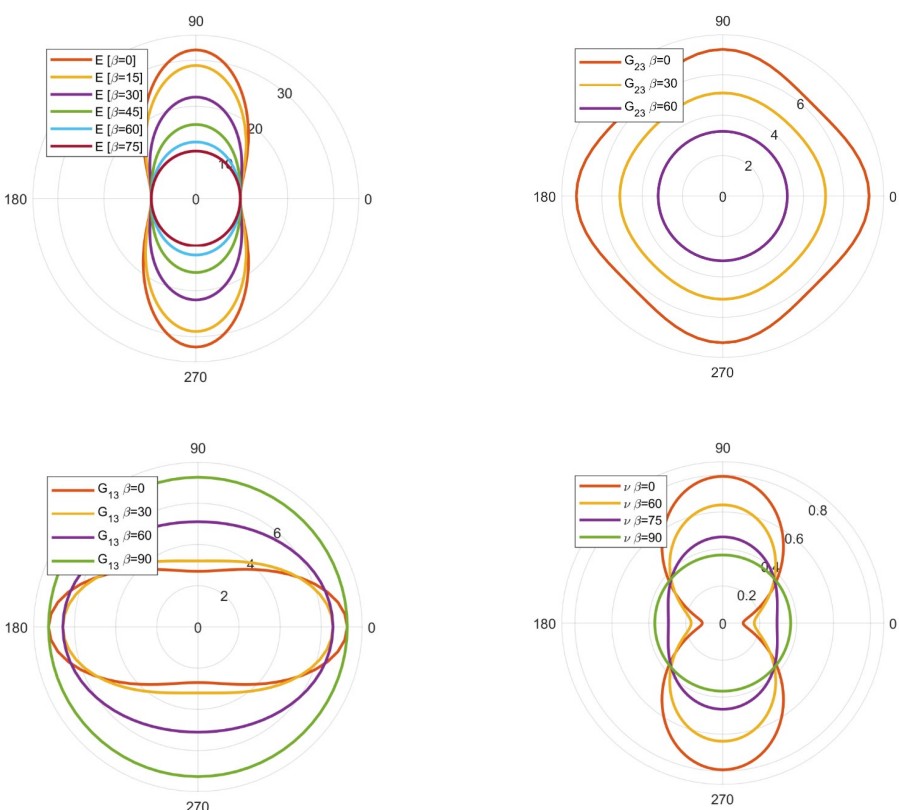

**Figure A6.** Benchmarks to Li (1976, their Figure 3) for hexagonal thallium.



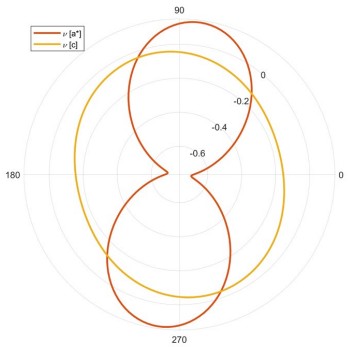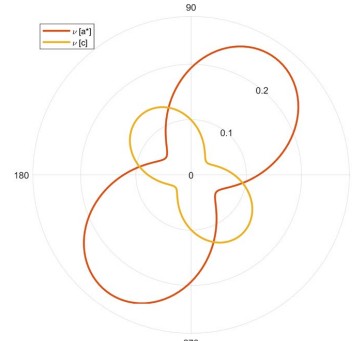

**Figure A7.** Benchmarks to Mainprice et al. (2008, their Figure 5) for triclinic talc (c1) at 0.0 GPa (left) and 3.9 GPa (right). The lower pressure example shows auxetic behaviour.

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
