# Peer review of "The variation and visualisation of elastic anisotropy in rock-forming minerals"

_Solid Earth, 2019_

## Referee Comment (RC1) · Marco Mercuri (Referee) · 9 Dec 2019

General comments:

The manuscript deals with the elastic anisotropies of minerals. More in detail, the authors present a new software/tool called AnisoVis that allows to investigate and to efficiently visualize the anisotropies of elastic moduli (and other parameters such as optical axes orientations and acoustic properties) of minerals. The new software is distributed with a very large collection of data taken from literature, and this allows a review of general and some specific (i.e., related to a particular mineral) aspects dealing with the elastic anisotropies of minerals. I recommend publication on Solid Earth after moderate revisions.

[Figure]

I think that, although the themes of the manuscript are interesting and innovative, the aim of the manuscript is not focussed enough. In detail, it is not clear if the authors are presenting (1) a review of the elastic moduli anisotropies in minerals and their consequences on solid earth problems, or (2) to present a new tool (AnisoVis) for the analysis and the visualization of elastic anisotropies.

Reading the present version of the manuscript I have the impression that the two topics mentioned above are presented separately. In my opinion, the manuscript should be more focused on the presentation of AnisoVis and on discussing its benefits for studies dealing with elastic anisotropies of minerals. At the same time, I think that an extensive review of literature data on elastic anisotropies, can be useful for the readers. My suggestion is to add, if possible, a (simple?) script in AnisoVis that allows to plot altogether the data for all the minerals (something similar to the Figures reported in section 4). In this way (1) the users of AnisoVis can completely benefit of the extensive collection of data, and (2) the manuscript can be easily rearranged by the authors by presenting the potential of AnisoVis to re-evaluate or reinterpret our knowledge of elastic anisotropies of minerals.

Finally, I think that the manuscript needs a discussion section where the benefits apported by AnisoVis and/or the novelty of the results reported in Section 4 and 5 are clearly stated.

Specific and technical comments:

1) Introduction In my opinion the Introduction is not focussed enough on the aim of the manuscript (see major comment). In this section the authors deal with many topics: the importance of the elastic anisotropies and their effects on common solid earth problems (seismic waves velocities, brittle and plastic behaviour etc..), the need for a more readable visualisation of the elastic properties, and the need for a review of the existing literature on elastic anisotropies. The result is a very long introduction in which the reader can lost himself quite easily. I suggest rearranging this section, focusing it

on the need of a software that is capable to (1) visualise (and hence interpret) in an efficient way the elastic anisotropies of minerals and, (2) review the existing literature.

Line 46-47: At least a reference should be provided for the sentence "these variations show a close relationship to the symmetry of the mineral crystallographic structure"

Lines 48, 50 and others: I think that a clear definition for "ab initio calculations" should be provided.

Lines 55-58: In my opinion the paper should be focussed on this concept. The Aniso-Vis software/tool brings 2 main advantages: 1) visualise in a high readable way and interpret the elastic anisotropies of minerals 2) It contains an extensive collection of mineral data

63-71: Dealing with the role of elastic anisotropies on the velocity of seismic waves, what is the knowledge gap the authors want to fill with their contribution?

2) Theory and underlying equations Line 191: I think that calling with "s" the compliance and with "c" the stiffness can confuse the reader. If possible, I suggest inverting the variables' names.

Line 205-206: Maybe it is worth mentioning the meaning of the first and second sub-scripts.

Line 265-270: I think that a more exhaustive explanation of the Universal Anisotropy Index is needed in order to fully understand the figures and the results (e.g. Figs. 11, 12, 17). For example: has the AU a range of values? What's the meaning of a high AU?

3) AnisoVis – program description and visualisation methods Lines 275-276: the link is missing Lines 326-327: It is clear how to produce representation surfaces, unit sphere and stereographic projections but I don't see in the GUI how to make a polar plot. Is it possible? If the answer is yes, it should be clearly described in the text. Furthermore, is it possible to select the plane of representation? Lines 340-343: I think that it would

be nice to develop in the future a script that allows to visualize the variation of the Poisson's ratio and Shear modulus within the surface that is orthogonal to the direction of application of the stress. Lines 340-343: Is it possible to retrieve the direction of the maximum and minimum shear modulus and Poisson's ratio? Lines 372-378. I expect (and I see from Fig. 12) that most of the maximum Poisson's ratios are positive. However, I think that the choice of not representing it should be clearly explained in the text. Lines 399-402: In my opinion, these lines are more appropriate for a "discussion" section. Lines 403-404 (section "Visualising second rank tensors: stress and strains). In my opinion this section does not add too much to the aim of the manuscript. I suggest removing this part. Lines 428-435 I suggest moving this part in the Methods section or in the next section.

4) Results – General trends In this section the general trends of elastic anisotropies are presented by plotting the various elastic moduli against the AU. The AU (Universal Elastic Anisotropy Index) has been introduced in Section 2 but very shortly (see the comment on Section 2). These plots are hence quite difficult to read. Moreover, if the author's aim it is to review the elastic anisotropies of minerals I think that it would be more appropriate to see the statistical distribution (using box plots or histograms) of the AU and/or the maximum/minimum values of the various elastic moduli and/or their difference from the Voigt-Reuss-Hill average. Finally, I think that, if the AU represents "how much" elastically anisotropic a mineral is, then it should be directly proportional to Emax/Emin (if the latter is a good indicator of anisotropy). Is it?

Lines 440-441: In Figure 10 is actually represented the Young's modulus vs. the AU. Lines 442-444: There are so many data points in Figure 10. In my opinion it is difficult to identify Emax and Emin for each mineral and do the math. I suggest to graphically render the distribution of Emax/Emin. Line 444: Actually, looking at Figure 10 it seems to me that Emax/Emin remains pretty constant for all the AU range. A figure where Emax/Emin is plotted against AU would help for this. Lines 444-445: see the comment above (lines 440-441) Lines 446-447: see the comment above (lines 442-444) Figure

10-13: The font is very small. Please make the writings bigger. Line 461: Figure 13a not 13 Figure 14: Please make the text bigger Lines 496-505 In order to produce a simpler plot, I suggest representing also the evolution of $\nu$max/$\nu$min with pressure and temperature. Figure 15b: data on Corundum are very difficult to see because of the yellow colour. Can you change it, please? Lines 511-515 and Fig. 17. Again, my suggestion is to represent $\beta$max/ $\beta$min. Please make the text in the figures bigger. Lines 529-535 and Fig. 18. I suggest explaining this representation of the elastic anisotropies somewhere in the text: what the dotted diagonal lines are?

5) Results – specific examples In this section, the authors rise some points on the possible implications of the elastic anisotropies (twinning, phase transformations, metamorphic reactions…) that need to be furtherly investigated. In my opinion, although the points raised are interesting, the role of AnisoVis for these results is not highlighted enough (an exception occurs for the section entitled "Brittle cracking, decrepitation and dehydration). My suggestion is to better explain the role of AnisoVis (if any) for the results or, alternatively, to shortly rise these points in the Introduction or Discussion section and remove this part.

---

## Referee Comment (RC2) · Anonymous Referee #2 · 9 Jan 2020

In this work, the authors explore the variation of the elastic anisotropy in rock forming minerals using previously published data and present a new open source software (AnisoVis) to compute and visualise that anisotropy. Finally, they remark the importance of the mineral elastic anisotropy for processes in the solid earth (e.g deformation, twinning, coherent phase transformations and brittle failure. The software is very simple to use also for inexpert users in programming languages. I think that the AnisoVis software is a powerful tool that can be used both in teaching and research environments. I recommend publication on Solid Earth.

———————————————

---

## Author Comment (AC1) · 20 Jan 2020

Referee #1 Marco Mercuri marco.mercuri@uniroma1.it

Authors responses labelled 'RESPONSE'.

General comments: The manuscript deals with the elastic anisotropies of minerals. More in detail, the authors present a new software/tool called AnisoVis that allows to investigate and to efficiently visualize the anisotropies of elastic moduli (and other parameters such as optical axes orientations and acoustic properties) of minerals. The new software is distributed with a very large collection of data taken from literature, and this allows a review of general and some specific (i.e., related to a particular mineral)

[Figure]

aspects dealing with the elastic anisotropies of minerals. I recommend publication on Solid Earth after moderate revisions.

I think that, although the themes of the manuscript are interesting and innovative, the aim of the manuscript is not focussed enough. In detail, it is not clear if the authors are presenting (1) a review of the elastic moduli anisotropies in minerals and their consequences on solid earth problems, or (2) to present a new tool (AnisoVis) for the analysis and the visualization of elastic anisotropies.

RESPONSE: This is a fair comment, and we struggled to find the right balance between (1) and (2). Our main intent is to focus on (1) – i.e. to assess and illustrate the variation in elastic properties both across rock-forming minerals and also within individual minerals (anisotropy). In order to deliver that, we developed the AnisoVis code (2). We have added new text to the Introduction to emphasise our intent.

Reading the present version of the manuscript I have the impression that the two topics mentioned above are presented separately. In my opinion, the manuscript should be more focused on the presentation of AnisoVis and on discussing its benefits for studies dealing with elastic anisotropies of minerals. At the same time, I think that an extensive review of literature data on elastic anisotropies, can be useful for the readers. My suggestion is to add, if possible, a (simple?) script in AnisoVis that allows to plot altogether the data for all the minerals (something similar to the Figures reported in section 4). In this way (1) the users of AnisoVis can completely benefit of the extensive collection of data, and (2) the manuscript can be easily rearranged by the authors by presenting the potential of AnisoVis to re-evaluate or reinterpret our knowledge of elastic anisotropies of minerals.

RESPONSE: These are also fair points. We have added new functionality to AnisoVis to produce the summary graphs reported in Section 4, and added new histograms for the variations in E, G and ïAć. As above, we have clarified our intent for the paper in terms of describing and illustrating the variation of elastic properties of rock-forming

minerals.

Finally, I think that the manuscript needs a discussion section where the benefits apported by AnisoVis and/or the novelty of the results reported in Section 4 and 5 are clearly stated.

RESPONSE: We have now included these statements in the Summary section.

Specific and technical comments:

1) Introduction

In my opinion the Introduction is not focussed enough on the aim of the manuscript (see major comment). In this section the authors deal with many topics: the importance of the elastic anisotropies and their effects on common solid earth problems (seismic waves velocities, brittle and plastic behaviour etc..), the need for a more readable visualisation of the elastic properties, and the need for a review of the existing literature on elastic anisotropies. The result is a very long introduction in which the reader can lost himself quite easily. I suggest rearranging this section, focusing it on the need of a software that is capable to (1) visualise (and hence interpret) in an efficient way the elastic anisotropies of minerals and, (2) review the existing literature.

RESPONSE: We slightly disagree (as above). In our opinion, the need for visualisation software (e.g. AnisoVis) is predicated on the importance of elastic anisotropy in rock-forming minerals, and it's relevance to solid earth problems. Therefore, we think it is reasonable to assess this importance and relevance in the Introduction and simultaneously acknowledge at least some of the vast literature in the field. The generation of AnisoVis has been a consequence of our desire to understand the range of natural variation in elastic properties, and their consequences, especially for deformation.

Line 46-47: At least a reference should be provided for the sentence "these variations show a close relationship to the symmetry of the mineral crystallographic structure"

RESPONSE: Added references.

Lines 48, 50 and others: I think that a clear definition for "ab initio calculations" should be provided.

RESPONSE: Added 'molecular dynamics'.

Lines 55-58: In my opinion the paper should be focussed on this concept. The Aniso-Vis software/tool brings 2 main advantages: 1) visualise in a high readable way and interpret the elastic anisotropies of minerals 2) It contains an extensive collection of mineral data

RESPONSE: We agree, although with a slightly different emphasis: for us, AnisoVis is a means to an end, and that 'end' is to assess, visualise and understand the variations within rock-forming minerals.

63-71: Dealing with the role of elastic anisotropies on the velocity of seismic waves, what is the knowledge gap the authors want to fill with their contribution?

RESPONSE: We do not aim to fill any specific knowledge gap in seismic (acoustic) velocities. We simply mention their intrinsic relationship to the underlying elastic properties, the focus of the paper.

2) Theory and underlying equations

Line 191: I think that calling with "s" the compliance and with "c" the stiffness can confuse the reader. If possible, I suggest inverting the variables' names.

RESPONSE: We agree this is confusing, but this is the standard nomenclature in standard texts and papers.

Line 205-206: Maybe it is worth mentioning the meaning of the first and second subscripts.

RESPONSE: The I and J subscripts can take the values 1, 2 or 3, and thereby link to the standard Cartesian reference frame (x = 1, y = 2, z = 3), as stated in the text.

Line 265-270: I think that a more exhaustive explanation of the Universal Anisotropy Index is needed in order to fully understand the figures and the results (e.g. Figs. 11, 12, 17). For example: has the AU a range of values? What's the meaning of a high AU?

RESPONSE: New text added to this section.

3) AnisoVis – program description and visualisation methods

Lines 275-276: the link is missing

RESPONSE: Added links to GitHub and MathWorks.

Lines 326-327: It is clear how to produce representation surfaces, unit sphere and stereographic projections but I don't see in the GUI how to make a polar plot. Is it possible? If the answer is yes, it should be clearly described in the text. Furthermore, is it possible to select the plane of representation?

RESPONSE: We have added polar 2D line plots to the AnisoVis GUI. At the present time, we have restricted these to 3 fixed planes (xy, xz and yz).

Lines 340-343: I think that it would be nice to develop in the future a script that allows to visualize the variation of the Poisson's ratio and Shear modulus within the surface that is orthogonal to the direction of application of the stress.

RESPONSE: We will add this to the list of future enhancements. This issue is non-trivial for both shear modulus (G) and Poisson's ratio (nu) because, as described in the text, these properties depend on two orthogonal directions (shear stress and shear strain in the case of G; and lateral strain and axial strain in the case of nu).

Lines 340-343: Is it possible to retrieve the direction of the maximum and minimum shear modulus and Poisson's ratio?

RESPONSE: These values are in fact displayed at the top of the plots for shear modulus and Poisson's ratio produced by AnisoVis (although not shown in the paper figures).

One direction is reported (in direction cosine format, [l, m, n]), although often there are many directions showing the same maximum (or minimum) due to the crystal symmetry. It is impractical to list all of them. For a future enhancement to AnisoVis we are working on translating these directions into the closest Miller indices.

Lines 372-378. I expect (and I see from Fig. 12) that most of the maximum Poisson's ratios are positive. However, I think that the choice of not representing it should be clearly explained in the text.

RESPONSE: Yes, text has been added.

Lines 399-402: In my opinion, these lines are more appropriate for a "discussion" section.

RESPONSE: We respectfully disagree, and think that these 2 sentences serve to round off the presentation of the different attributes and plot formats.

Lines 403-404 (section "Visualising second rank tensors: stress and strains). In my opinion this section does not add too much to the aim of the manuscript. I suggest removing this part.

RESPONSE: We rely on these display formats in our subsequent discussion of strain energy in quartz, and think that they complement the previous plots. We prefer to retain them as is.

Lines 428-435 I suggest moving this part in the Methods section or in the next section.

RESPONSE: Agreed. We have moved this into the next section.

4) Results – General trends

In this section the general trends of elastic anisotropies are presented by plotting the various elastic moduli against the AU. The AU (Universal Elastic Anisotropy Index) has been introduced in Section 2 but very shortly (see the comment on Section 2). These plots are hence quite difficult to read. Moreover, if the author's aim it is to

review the elastic anisotropies of minerals I think that it would be more appropriate to see the statistical distribution (using box plots or histograms) of the AU and/or the maximum/minimum values of the various elastic moduli and/or their difference from the Voigt-Reuss-Hill average.

RESPONSE: These are good points. We have added plots for the ratio of Emax/Emin etc., and now include plots for the statistical distributions of E, G and beta (in addition to those for Poisson's ratio already supplied). These are also available to the user of AnisoVis through the new 'Summary plots' button.

Finally, I think that, if the AU represents "how much" elastically anisotropic a mineral is, then it should be directly proportional to Emax/Emin (if the latter is a good indicator of anisotropy). Is it?

RESPONSE: Sort of! See new Figure 10b.

Lines 440-441: In Figure 10 is actually represented the Young's modulus vs. the AU.

RESPONSE: We disagree; by showing Emax, Emin and EVRH we are illustrating one aspect of the anisotropy of each mineral. No change to the text.

Lines 442-444: There are so many data points in Figure 10. In my opinion it is difficult to identify Emax and Emin for each mineral and do the math. I suggest to graphically render the distribution of Emax/Emin.

RESPONSE: Agreed. Done.

Line 444: Actually, looking at Figure 10 it seems to me that Emax/Emin remains pretty constant for all the AU range. A figure where Emax/Emin is plotted against AU would help for this.

RESPONSE: As above, done. And not that constant! Interesting plot.

Lines 444-445: see the comment above (lines 440-441)

[Figure]

RESPONSE: Similar plot for G implemented.

Lines 446-447: see the comment above (lines 442-444)

RESPONSE: Similar plots for nu and beta implemented.

Figure 10-13: The font is very small. Please make the writings bigger.

RESPONSE: Done.

Line 461: Figure 13a not 13

RESPONSE: Done.

Figure 14: Please make the text bigger

RESPONSE: Done.

Lines 496-505 In order to produce a simpler plot, I suggest representing also the evolution of _max/_min with pressure and temperature.

RESPONSE: Done.

Figure 15b: data on Corundum are very difficult to see because of the yellow colour. Can you change it, please?

RESPONSE: Done.

Lines 511-515 and Fig. 17. Again, my suggestion is to represent _max/ _min. Please make the text in the figures bigger.

RESPONSE: Done.

Lines 529-535 and Fig. 18. I suggest explaining this representation of the elastic anisotropies somewhere in the text: what the dotted diagonal lines are?

RESPONSE: Text added.

5) Results – specific examples

In this section, the authors rise some points on the possible implications of the elastic anisotropies (twinning, phase transformations, metamorphic reactions: : :) that need to be furtherly investigated. In my opinion, although the points raised are interesting, the role of AnisoVis for these results is not highlighted enough (an exception occurs for the section entitled "Brittle cracking, decrepitation and dehydration). My suggestion is to better explain the role of AnisoVis (if any) for the results or, alternatively, to shortly rise these points in the Introduction or Discussion section and remove this part.

RESPONSE: As our focus is actually on the variation of elastic properties, and only in passing on AnisoVis, we think the overall content here is fine. As noted though, outputs from AnisoVis are useful, and we have used these in other papers too (Timms et al., 2018).

RESPONSE: Finally, we'd like to thank the reviewer again for taking the time to provide us with such useful and detailed comments. We think the changes to the manuscript and code have led to a significant improvement.

―――――――――――――――――

---

## Author Comment (AC2) · 20 Jan 2020

Referee #2 Anonymous

Authors responses labelled 'RESPONSE:'.

In this work, the authors explore the variation of the elastic anisotropy in rock forming minerals using previously published data and present a new open source software (AnisoVis) to compute and visualise that anisotropy. Finally, they remark the importance of the mineral elastic anisotropy for processes in the solid earth (e.g. deformation, twinning, coherent phase transformations and brittle failure. The software is very simple to use also for inexpert users in programming languages. I think that the AnisoVis software is a powerful tool that can be used both in teaching and research

environments. I recommend publication on Solid Earth.

RESPONSE: We thank the referee for their positive review. We also hope that Aniso-Vis will be used for research and teaching. We have not made any changes to the manuscript in response to this review.